# Oxidized mitochondrial DNA induces gasdermin D oligomerization in systemic lupus erythematosus

Naijun Miao [1,8], Zhuning Wang [1,8], Qinlan Wang[1], Hongyan Xie[2], Ninghao Yang[2], Yanzhe Wang[3], Jin Wang[1], Haixia Kang[1], Wenjuan Bai[1], Yuanyuan Wang[1], Rui He[1], Kepeng Yan[1], Yang Wang[1], Qiongyi Hu [4], Zhaoyuan Liu [1], Fubin Li [1], Feng Wang[1], Florent Ginhoux [5], Xiaoling Zhang [6], Jianyong Yin [7]✉, Limin Lu [2]✉ & Jing Wang [1]✉

Although extracellular DNA is known to form immune complexes (ICs) with autoantibodies in systemic lupus erythematosus (SLE), the mechanisms leading to the release of DNA from cells remain poorly characterized. Here, we show that the pore-forming protein, gasdermin D (GSDMD), is required for nuclear DNA and mitochondrial DNA (mtDNA) release from neutrophils and lytic cell death following ex vivo stimulation with serum from patients with SLE and IFN-γ. Mechanistically, the activation of FcγR downregulated Serpinb1 following ex vivo stimulation with serum from patients with SLE, leading to spontaneous activation of both caspase-1/caspase-11 and cleavage of GSDMD into GSDMD-N. Furthermore, mtDNA oxidization promoted GSDMD-N oligomerization and cell death. In addition, GSDMD, but not peptidyl arginine deiminase 4 is necessary for extracellular mtDNA release from low-density granulocytes from SLE patients or healthy human neutrophils following incubation with ICs. Using the pristane-induced lupus model, we show that disease severity is significantly reduced in mice with neutrophil-specific *Gsdmd* deficiency or following treatment with the GSDMD inhibitor, disulfiram. Altogether, our study highlights an important role for oxidized mtDNA in inducing GSDMD oligomerization and pore formation. These findings also suggest that GSDMD might represent a possible therapeutic target in SLE.

Systemic lupus erythematosus (SLE) is a heterogeneous autoimmune disease characterized by the breakdown of tolerance against nucleic acids, leading to systemic damage to peripheral organs[1–3]. Dysregulation of apoptosis has been proposed to have a key role in tolerance failure, activation of autoimmune lymphocytes and tissue damage in SLE[4,5]. In addition to apoptosis, necrotic cell death is also important in the initiation and perpetuation of SLE[5]. However, the involvement pyroptosis, a gasdermin-mediated

[1]Center for Immune-related Diseases at Shanghai Institute of Immunology, Ruijin Hospital, Shanghai Jiao Tong University School of Medicine, Shanghai, China. [2]Department of Physiology and Pathophysiology, School of Basic Medicine Sciences, Fudan University, Shanghai, China. [3]Department of Nephrology, Shanghai Tong Ren Hospital, Shanghai Jiao Tong University School of Medicine, Shanghai, China. [4]Department of Rheumatology and Immunology, Ruijin Hospital, Shanghai Jiao Tong University School of Medicine, Shanghai, China. [5]Singapore Immunology Network, Agency for Science, Technology and Research, Singapore, Singapore. [6]Department of Orthopedic Surgery, Xinhua Hospital, Shanghai Jiao Tong University School of Medicine, Shanghai, China. [7]Department of Nephrology, Shanghai Jiao Tong University Affiliated Sixth People's Hospital, Shanghai, China. [8]These authors contributed equally: Naijun Miao, Zhuning Wang. ✉e-mail: yinjianyong09@163.com; lulimin@shmu.edu.cn; jingwang@shsmu.edu.cn

form of programmed necrotic cell death[6], remains unconfirmed in SLE.

Gasdermin D (GSDMD) is a pore-forming protein that plays a key role in pyroptosis[7,8]. Following its cleavage by caspase-1 or caspase-11, GSDMD-N translocates to the plasma membrane where it oligo-merizes to form membrane pores, resulting in cellular lysis and inflammatory cytokine release[9,10]. In addition to permitting the release of pro-inflammatory cytokines, GSDMD pores can also facil-itate the release of tissue factor and lectin, thereby to further promote inflammation[11,12]. Despite the importance of GSDMD in pyroptosis, mechanisms regulating its activity are largely focused on upstream factors that regulate its cleavage, such as NLRP3 inflammasome, caspase-11 or caspase-8[13,14]. Whether other factors promote pore for-mation at the plasma membrane after GSDMD cleavage is less clear. A recent study identified that Ragulator-Rag promotes GSDMD oli-gomerization through mitochondrial reactive oxygen species (mROS) in macrophages[15]. However, the precise mechanism through which mROS regulate GSDMD oligomerization remains poorly described.

Mitochondrial DNA (mtDNA) is a multi-copy, circular, double-stranded DNA molecule that is essential for oxidative phosphorylation (OXPHOS)[16,17]. Dysregulation of mtDNA has been associated with mul-tiple diseases including neurodegenerative disorders, metabolic dis-ease, heart failure and cancer[18-20]. In SLE, neutrophil mtDNA is highly accessible to be oxidized by mROS and released from mitochondria[21,22]. Cytosolic mtDNA further promotes type I interferon (IFN) production through the cGAS-STING pathway[23,24]. In addition, mtDNA can also be released into the extracellular environment[25]. Extracellular mtDNA activates plasmacytoid dendritic cells and initiates CD4+ T cell activa-tion, which is critical to SLE pathogenesis[21,26]. Recently, mtDNA was identified as an intracellular second messenger under genotoxic stress[27]. However, the role of mtDNA after mROS stress in SLE is incompletely understood.

In the present study, we provide evidence that GSDMD activation in neutrophils strongly correlates with plasma mtDNA levels and dis-ease activity in SLE patients. Moreover, we further uncover a novel mechanism of oxidized mtDNA (Ox-mtDNA)-dependent GSDMD oli-gomerization via a direct interaction with the Gasdermin D-N (GSDMD-N) terminal. We also evaluate the therapeutic potential of targeting GSDMD in the context of SLE.

## Results

### GSDMD is activated in neutrophils of SLE patients and lupus mice

Damage associated molecular patterns (DAMPs) which can be detec-ted by innate immune receptors are important regulators of auto-immune disease[28]. As the kidney is the most vulnerable organ in SLE, we measured the expression of DAMP receptors in kidneys from pristane-induced-lupus (PIL) mice, including Toll-like receptors (TLRs), NOD-like receptors (NLRs), the retinoic acid-inducible gene I (RIG-I), C-type lectin receptors (CLRs), intracellular DNA sensors, and asso-ciated downstream signaling molecules. We then focused on GSDMD, which was significantly upregulated together with caspase-1 and caspase-4 (also known as caspase-11 in mice) upregulation in lupus mice (Fig. 1a). We also detected a significant increase in IRF3 and IRF7 in PIL mice compared with control. Immunofluorescence of renal biopsy samples from lupus nephritis (LN) patients showed increased neutrophil infiltration in the glomeruli and tubulointerstitial, wherein these neutrophils exhibited higher levels of GSDMD expression as compared to the resident renal cells (Fig. 1b–d). Increased glomerular neutrophil infiltration was also detected in two murine lupus models, wherein the infiltrated neutrophils showed increased GSDMD expres-sion as compared to the control (Fig. 1e–i). The activation of GSDMD requires cleavage of upstream protease-dependent full-length GSDMD into the GSDMD-N terminal[10]. The cleaved GSDMD-N is detected only in the bone marrow neutrophils from PIL or MRL/lpr mice (Fig. 1j, k).

GSDMD is mainly localized to the plasma membrane in neutrophils from PIL or MRL/lpr mice (Supplementary Fig. 1a–d).

Similarly, flow cytometry analysis revealed cell surface expression of GSDMD in peripheral blood neutrophils from SLE patients but not healthy volunteer (HV) (Fig. 1l, m). There was also increased cell surface expression of GSDMD on neutrophils from PIL or MRL/lpr mice com-pared with disease-free animals (Supplementary Fig. 1e, f). However, there was no difference of GSDMD expression on monocytes from MRL/lpr and PIL mice compared with disease-free animals (Supple-mentary Fig. 1g, h). The mRNA expression of GSDMD in peripheral blood neutrophils from SLE patients was increased compared with HV (Fig. 1n). The level of full-length GSDMD expression was increased compared with HV, whereas GSDMD-N was only detected in SLE patients (Fig. 1o, p). GSDMD is also translocated to the plasma mem-brane in peripheral blood neutrophils from SLE patients (Supple-mentary Fig. 1i, j). The expression of GSDMD-N in neutrophils correlated positively with the SLE disease activity index (SLEDAI) (Fig. 1q). GSDMD plays a vital role in the generation of NETs[29,30]. Indeed, GSDMD-N expression correlated positively with the level of Myeloperoxidase (MPO)-DNA complexes (Fig. 1r) and elastase-DNA complexes (Fig. 1s), thus indicating a relationship between GSDMD activation and NETs. Recently, mtDNA in the serum of SLE patients has been implicated in disease pathogenesis[25]. The expression of GSDMD-N from SLE patients correlated positively with serum mtDNA levels (Fig. 1t). In addition, a significant increase in propidium iodide (PI) staining was observed in neutrophils from SLE patients compared with HV (Supplementary Fig. 2a, b). Collectively, these results suggested that GSDMD was significantly activated in neutrophils from both lupus mice and SLE patients and may contribute to the generation of NETs and mtDNA release.

### GSDMD-dependent neutrophil death in kidneys of lupus-like mice

GSDMD is a membrane pore-forming protein, which plays an essential role in pyroptosis[7,8]. We performed two photon intravital imaging to track neutrophil cell death in the kidneys of lupus-like mice. Rapid capture and accumulation of neutrophils within the glomerular capil-laries were observed in both control and pristane-induced-lupus (PIL) mice (Supplementary Movies 1 and 2). Neutrophil infiltration increased significantly in the glomeruli and tubulointerstitial tissues of PIL mice (Fig. 2a, b). We then used granulocyte-monocyte progenitor Ms4a3-tdTomato reporter mice, in which almost 100% of the neutrophils are labeled with tdTomato[31]. Immunofluorescence staining of the kidneys of pristane-treated Ms4a3 mice confirmed that approximately 90% of the infiltrating tdTomato+ cells were neutrophils (Fig. 2c, d). Following Sytox Green administration, Sytox Green-stained DNA release was found from tdTomato+ cells in the form of nuclear split and subsequent extracellular release (Fig. 2e and Supplementary Movie 3) or net-like structures (Fig. 2f and Supplementary Movies 4) were observed. Some NETs were resistant to high shear conditions and present for pro-longed periods in the glomerular capillaries (-10 min as punctate structures and >30 min as mesh-like structures). Furthermore, pristine-induced Sytox Green positive staining neutrophils in renal sections reduced significantly in Gsdmd−/− lupus mice (Fig. 2g and Supplemen-tary Movies 5–6), thus indicating GSDMD-dependent cell death in lupus mice. These results indicated GSDMD-dependent neutrophil extracellular DNA release in lupus mice.

### GSDMD pores permit DNA release from lupus neutrophils
To confirm whether GSDMD was required for the release of extra-cellular DNA from neutrophils in SLE, we cultured neutrophils from HV with serum from SLE patients. Caspase-1, caspase-11 and GSDMD were upregulated by stimulation with interferon-γ (IFN-γ), via the upregu-lation of IFN-stimulated response elements (ISRE) and IFN-stimulated gene (ISG)[32,33]. Therefore, we used pretreatment with IFN-γ to prime

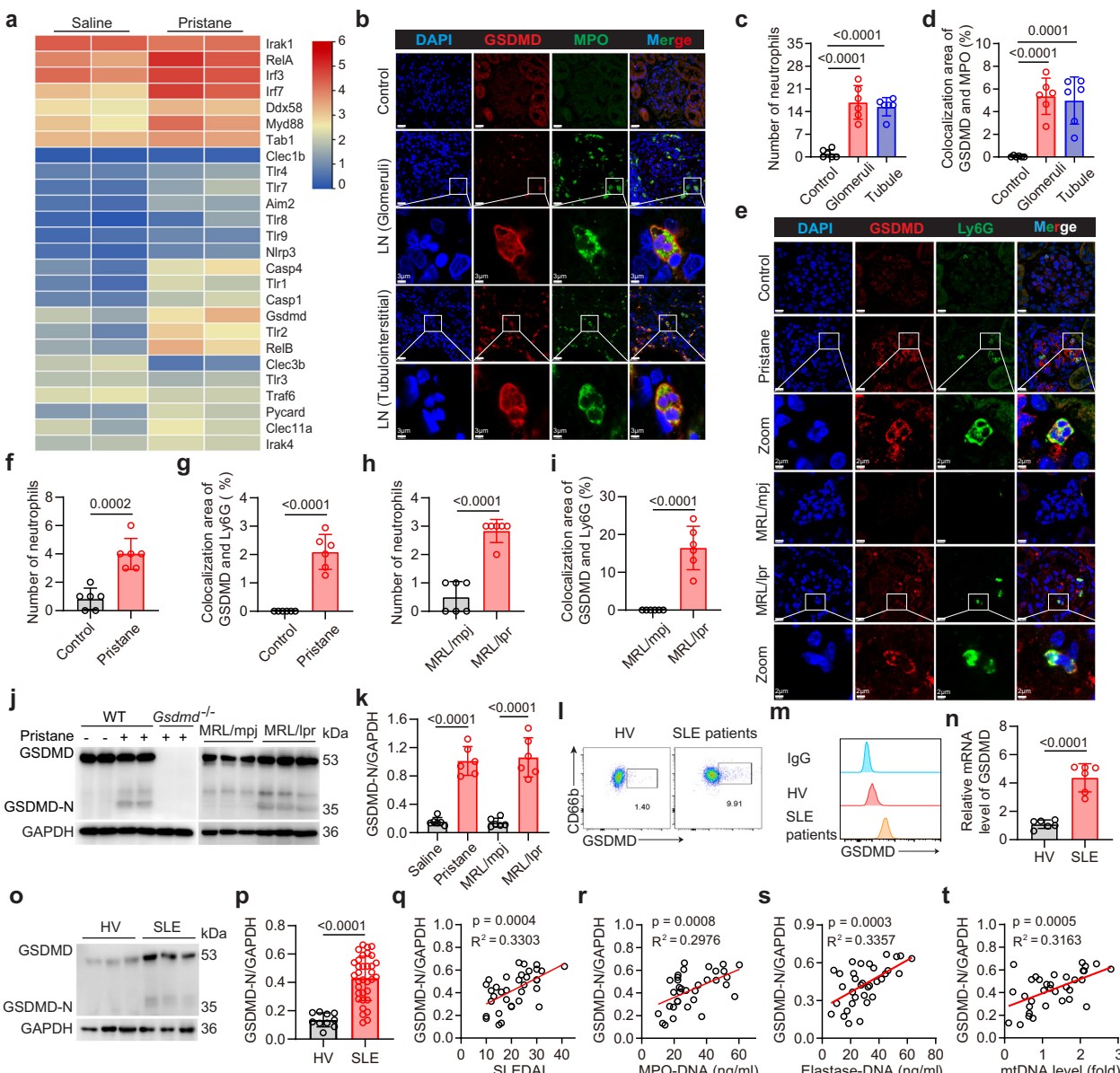

**Fig. 1 | GSDMD is activated in neutrophils from lupus mice and SLE patients.**
**a** Heatmap of genes involved in DAMP sensing pathways that are differentially expressed between the kidneys of mice treated with saline and pristane.
**b** Immunofluorescence staining for GSDMD and MPO in glomeruli and tubulointerstitial of renal biopsy from lupus nephritis (LN) patients. Scale bar, 20 μm, 3 μm (enlarged). **c, d** Quantitative analysis of numbers of neutrophils per field of view (FOV) or co-localization area of GSDMD and MPO in FOV. Tumor-adjacent normal tissue samples of 3 renal carcinoma patients or renal biopsies from 3 LN patients. The results are pooled from two independent experiments. **e** Immunofluorescence staining for GSDMD and Ly6G in kidney from PIL and MRL/lpr mice. Scale bar, 10 μm. 2 μm (enlarged). **f–i** Numbers of neutrophils or co-localization area of GSDMD and Ly6G per FOV. $n = 6$ mice. **j** Immunoblotting of GSDMD and GSDMD-N in bone marrow (BM) neutrophils from wide-type (WT), PIL, pristane-treated $Gsdmd^{-/-}$, MRL/mpj and MRL/lpr mice. **k** Quantitative analysis of GSDMD-N/GAPDH. $n = 6$ mice. The samples shown are from the same experiment. Three blots (PIL group)

and two blots (MRL/lpr group) were processed in parallel. **l** Flow cytometry plots for GSDMD in peripheral blood neutrophils from HV and SLE patients.
**m** Fluorescence intensity of GSDMD from the isotype, HV and SLE groups. **n** mRNA levels of GSDMD in HV and SLE patients. $n = 6$ HVs or SLE patients.
**o** Immunoblotting for GSDMD and GSDMD-N protein in peripheral blood neutrophils from HV and SLE patients. **p** Quantitative analysis of GSDMD-N/GAPDH. $n = 10$ HVs and $n = 34$ SLE patients. The samples shown are from the same experiment. Three blots were processed in parallel. Correlation of relative expression of GSDMD-N (the ratio of GSDMD-N to GAPDH from immunoblotting analysis) on neutrophils from SLE patients with SLEDAI (**q**), MPO-DNA complexes (**r**) elastase-DNA complexes (**s**) and mtDNA levels (**t**) in serum. Data are representative of two (**b**) or three (**a, e, l, m, n**) independent experiments. Data are presented as mean ± SD. Significance was examined by unpaired two-sided Student's $t$ test (**f–i, k, n, p**) or one-way ANOVA (**c, d**). Spearman's nonparametric test for (**q–t**).

neutrophils to increase the precursors of caspase-1, caspase-11 and GSDMD. Extensive neutrophil extracellular DNA release (as detected by Sytox Green staining) was observed following lupus serum (LS) treatment, and this release was suppressed upon treatment with the GSDMD inhibitor, disulfiram (DSF) (Fig. 3a, b). DSF treatment had no effect on the cell viability of bone marrow neutrophils and human

peripheral blood neutrophils (Supplementary Fig. 2e, f). The extracellular DNA was decorated with cit-H3 and elastase, thus indicating the formation of NETs. Similarly, the release of extracellular DNA from mouse bone marrow neutrophils was suppressed in $Gsdmd^{-/-}$ mice as compared to the wild-type (WT) mice following LS plus IFN-γ treatment (Fig. 3c, d). We quantified the DNA release into cell culture

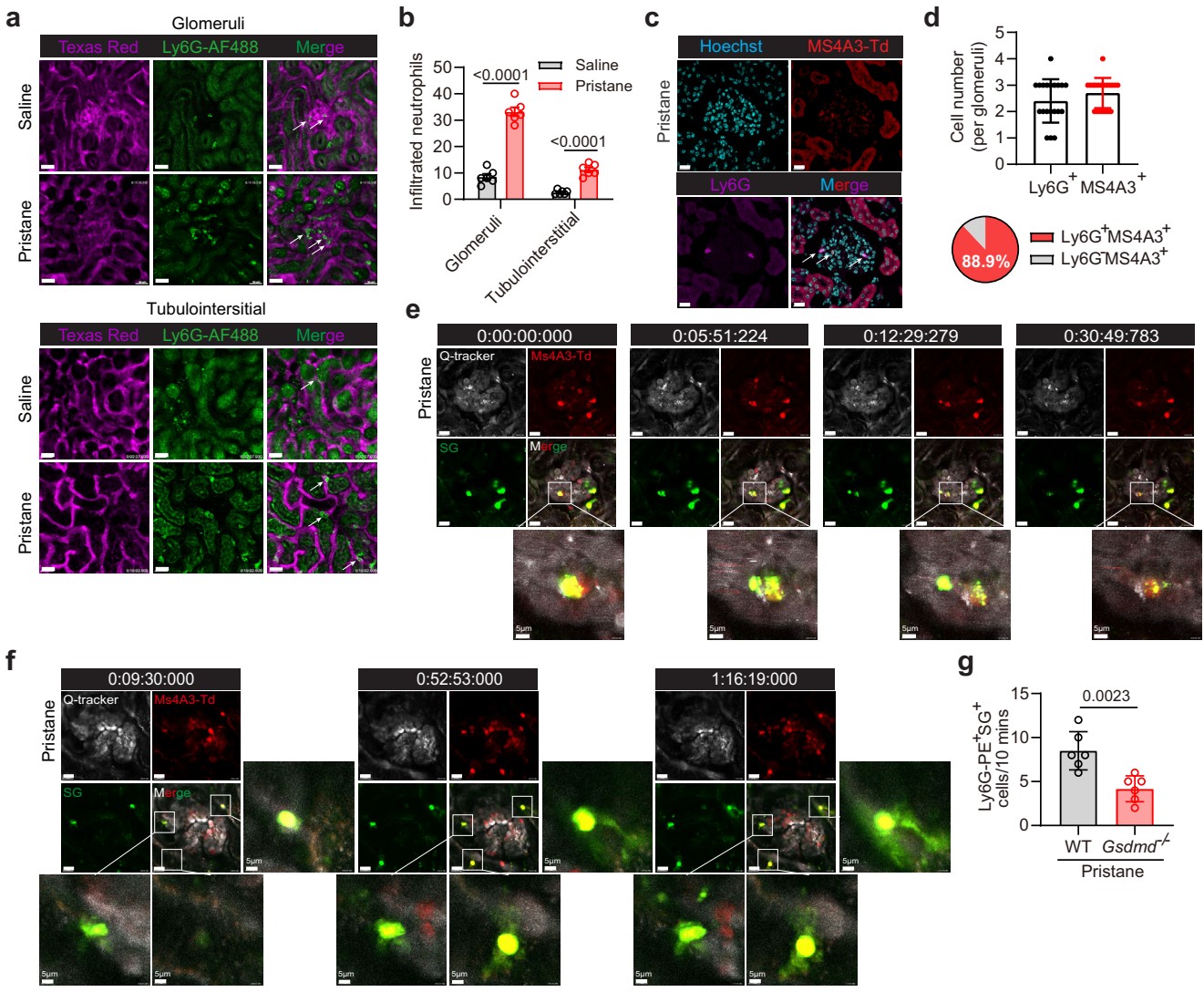

**Fig. 2 | Intravital renal microscopy reveals neutrophil cell death in live lupus mice. a** Two photon intravital imaging analysis of infiltrated neutrophils (Ly6G-AF488) in renal blood vessels (Texas Red) of live mice (glomeruli and tubulointerstitial) after saline or pristane treatment. Scale bar, 30 μm. **b** Quantitative analysis of infiltrated neutrophils in glomeruli and tubulointerstitial of mice after saline or pristane treatment. n = 6 mice. **c** Immunofluorescence staining of Ly6G in glomeruli from *Ms4a3-tdTomato* (*Ms4a3-Td*) mice after pristane treatment. Scale bar, 30 μm. **d** Quantitative analysis of Ly6G⁺ and Ms4a3-tdTomato⁺ cell numbers in each glomerulus and percentage of Ms4a3-tdTomato⁺ Cells. Representative of three independent experiments, and each point represents one glomerulus with

the mean being represented by a horizontal line. **e, f** Intravital imaging of the kidneys from pristane-treated *Ms4a3-Td* mice revealing the release of DNA (Sytox Green, green) from tdTomato⁺ cells (Red). Time-lapse images are shown of DNA that was released in the form of punctate particles (**e**) or mesh-like structures (**f**). Scale bar, 20 μm. 5 μm (enlarged). Representative of two independent experiments. **g** Quantitative analysis of PE-Ly6G⁺SG⁺ cells/10 min in kidney of PIL and *Gsdmd⁻/⁻* mice by intravital imaging. n = 6 mice. Representative of three independent experiments. Data are presented as mean ± SD. Significance was examined with unpaired two-sided Student's t test (**b, d, g**).

supernatants following a previously published method[34]. Neutrophil extracellular DNA release following lupus serum or PMA treatment decreased significantly in cells from *Gsdmd⁻/⁻* mice as compared to those from the WT mice (Fig. 3e). We also observed a reduced PI staining in *Gsdmd⁻/⁻* neutrophils compared with WT neutrophils after LS plus IFN-γ treatment (Supplementary Fig. 2c, d). These data suggest that GSDMD was essential for NET-associated DNA release after lupus serum treatment.

To explore the source of NET-associated DNA, we measured the levels of mtDNA and nuclear DNA by detecting the expression of their encoded genes in the supernatant after LS plus IFN-γ treatment. Both mtDNA and nuclear DNA in supernatant increased over time after lupus serum plus IFN-γ treatment. GSDMD knockout completely abolished the release of both mtDNA and nuclear DNA (Fig. 3f–i). Glycine can be used to prevent membrane rupture and lytic cell death

but not the formation of GSDMD pores[35]. Glycine treatment efficiently prevented cell lysis as measured by the release of LDH and nuclear DNA (Fig. 3h–j). However, glycine treatment showed no significant effects on LS plus IFN-γ-induced extracellular mtDNA release (Fig. 3f, g).

mtDNA is first released from mitochondria when cells are under stress including oxidative stress[36]. We examined the mitochondrial membrane potential in bone marrow neutrophils isolated from PIL or pristane-induced *Gsdmd⁻/⁻* mice. Most control neutrophils exhibited high mitochondrial membrane potential. Bone marrow neutrophils from PIL mice showed reduced mitochondrial membrane potential, whereas those from *Gsdmd⁻/⁻* mice showed a reduction in the percentage of hyperpolarized mitochondria (Fig. 3k, l). Mitochondrial reactive oxygen species (mROS) is a potential source of mitochondrial damage and has been implicated in the pathogenesis of SLE[23].

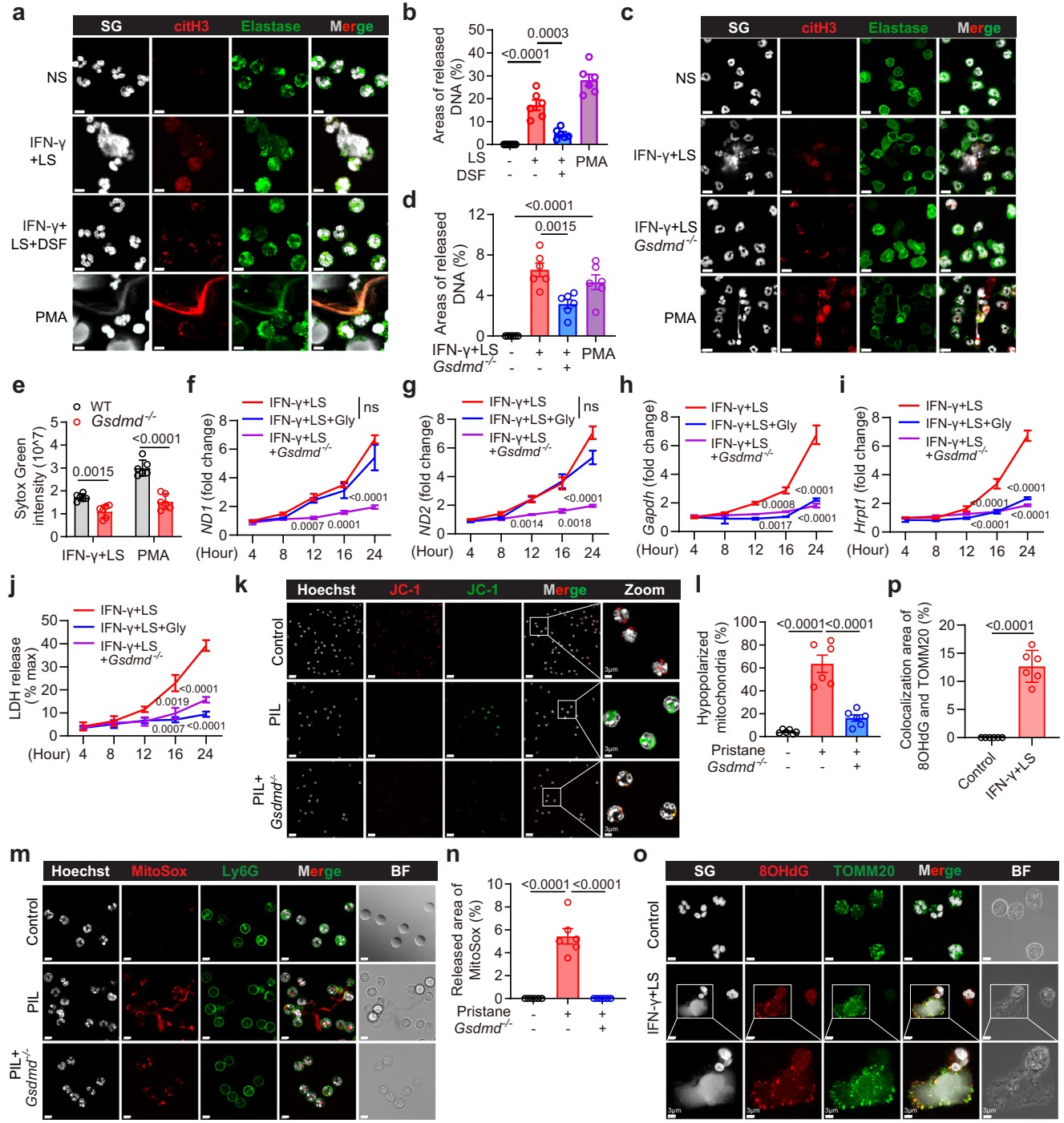

**Fig. 3 | GSDMD is required for the release of NET-associated DNA and mtDNA following lupus serum treatment. a** Immunofluorescence staining of citH3, elastase, and Sytox Green (SG). Cells were treated with normal serum (NS) from HV, LS from SLE patients plus IFN-γ, pretreated with DSF, or PMA. Scale bar, 5 μm. **b** Quantitative analysis of areas of released DNA in (**a**). Symbols represent the percentage of extracellular DNA area as compared to the entire FOV. Two healthy donors were used in one experiment, plots were pooled from three independent experiments using cells from 6 healthy donors. **c** Immunofluorescence staining of SG, citH3 and Ly6G. scale bar, 15 μm. **d** Quantitative analysis of areas of released DNA in (**c**). Symbols represent the area percentage of extracellular DNA relative to the entire FOV. **e** Quantitative analysis of SG intensity in the supernatant collected from WT and *Gsdmd*⁻/⁻ BM neutrophils following stimulation with LS plus IFN-γ or PMA. qPCR analysis of mitochondrial-encoded gene *ND1* (**f**) and *ND2* (**g**), nuclear-encoded gene, *Gapdh* (**h**) and *Hrpt1* (**i**) in the supernatant. *n* = 3 mice.

**j** Quantification of LDH in the supernatant from the indicated treatment groups. *n* = 3 mice. **k** Immunofluorescence staining of JC-1 in neutrophils from WT, PIL mice, and *Gsdmd*⁻/⁻ mice after pristane treatment. Scale bar, 20 μm. 3 μm (enlarged). **l** Quantification of hypopolarized mitochondria in (**k**). *n* = 6 mice. **m** Immunofluorescence staining of MitoSox and Ly6G from WT, PIL mice, and *Gsdmd*⁻/⁻ mice after pristane treatment. Scale bar, 5 μm. **n** Quantitative analysis of areas of released MitoSox in (**m**). *n* = 6 mice. **o** Immunofluorescence staining of SG, 8OHdG, and TOMM20 in neutrophils stimulated with LS + IFN-γ for 8 h; scale bar, 5 μm. 3 μm (enlarged). **p** Quantification of co-localization area of 8OHdG and TOMM20 in **o**. The results are pooled from two independent experiments using cells from 6 mice (**d**, **e**, **p**). Three independent experiments (**f–j**, **k–n**). Data are shown as mean ± SD. Significance was examined by one-way ANOVA (**b**, **d**, **l**, **n**), unpaired two-sided Student's *t* test (**e–j**, **p**).

Mitochondrial superoxide production increased in neutrophils from PIL mice as compared to those from control or *Gsdmd*[-/-] PIL mice, as determined by MitoSox staining (Fig. 3m, n). ROS can induce the oxidation of mtDNA. Indeed, using anti-8-Oxo-2′-deoxyguanosine (8OHdG) antibody to detect DNA oxidation, strong 8OHdG staining was detected both on neutrophils and extruded NETs following LS plus IFN-γ treatment (Fig. 3o, p). 8OHdG staining co-localized with trans-locase of outer mitochondrial membrane 20 (TOMM20) staining, a marker of mitochondria. Taken together, these data showed that GSDMD was required for lytic NETs and non-lytic oxidized mtDNA (Ox-mtDNA) release.

## GSDMD and mROS inhibitor suppresses DNA release from human neutrophils

Ribonucleoprotein immune complexes (RNP ICs) are prevalent in lupus and induce extracellular DNA release from neutrophils[22]. Indeed, immunofluorescent staining revealed the release of neutrophil extra-cellular DNA (as detected by Sytox Green staining), as well as Ox-mtDNA (as detected by TOMM20 and 8OHdG staining) following RNP ICs treatment; extracellular DNA release was suppressed upon treatment with DSF, GSK484 (PAD4 inhibitor), and Mito-TEMPO (Fig. 4a, b). However, the release of Ox-mtDNA was suppressed by DSF and Mito-TEMPO, but not GSK484 (Fig. 4a, b). Increased 8OHdG levels indicated that DNA oxidation and RNP ICs-dependent 8OHdG release in the culture medium were suppressed by DSF and Mito-TEMPO but not by GSK484 (Fig. 4c). RNP ICs-induced release of mtDNA in culture media was statistically decreased upon DSF and Mito-TEMPO treatment, whereas GSK484 treatment exhibited no effect (Fig. 4d). Low-density granulocytes (LDGs) are a distinct neutrophil subset found in individuals with SLE. They contribute to the pathogenesis of SLE through the release of extracellular DNA. LDGs spontaneously released extra-cellular DNA and Ox-mtDNA following in vitro culture for 6 h (Fig. 4e). DSF and Mito-TEMPO statistically suppressed the release of extra-cellular DNA and mtDNA from these cells, whereas the release was unaffected upon GSK484 treatment (Fig. 4f). Levels of 8OHdG and mtDNA in LDG culture media were markedly suppressed by DSF and Mito-TEMPO but not upon GSK484 treatment (Fig. 4g, h). In line with the above findings, treatment with DSF and Mito-TEMPO but not GSK484 was sufficient to protect neutrophils against RNP ICs-induced cell death as measured by LDH release (Fig. 4i). Quantification of LDH levels in the culture medium also confirmed that both DSF and Mito-TEMPO but not GSK484 protected LDGs from cell death (Fig. 4j). Collectively, these results showed that mROS and GSDMD were required for Ox-mtDNA release in RNP ICs stimulated human neu-trophils and LDGs from individuals with SLE.

## GSDMD-N is activated via the FcγR/Serpinb1/caspase-1/11 pathway

Activation of GSDMD has been identified in many non-infectious diseases[37–39]. However, the mechanisms that regulate the cleavage of GSDMD in neutrophils of lupus mice remain unknown. Bone marrow neutrophils were isolated from WT mice. Cleaved GSDMD-N terminal was only detected following LS plus IFN-γ stimulation (Fig. 5a, b). In addition, cleaved GSDMD-N was detected in neutrophils from human peripheral blood after LS treatment (Fig. 5c, d). The cleavage of GSDMD is mediated by the pyroptotic caspase-1, −4, and −5 in humans, and caspase-1 and −11 in mice[40]. We further examined the role of cas-pase-1/11 in LS-induced GSDMD cleavage. Single knockout of caspase-1 or caspase-11 was sufficient to partially attenuate the cleavage of GSDMD following LS plus IFN-γ treatment, whereas GSDMD-N was hardly detected in the caspase-1/11 double-knockout mice (Fig. 5e, f), indicating an essential role of both caspase-1 and −11 in GSDMD cleavage. Immune complexes (ICs) are the most abundant compo-nents in the serum of SLE patients and contribute to the activation of neutrophils and NETs[41]. IgG-depleted lupus serum failed to induce

increased activation of caspase-1 and −11, along with the cleavage of GSDMD in neutrophils (Fig. 5g, h). In addition, knocking out FcγR suppressed the activation of caspase-1 and −11, and the cleavage of GSDMD after LS plus IFN-γ stimulation (Fig. 5i, j). Previous studies suggest that a protease inhibitor, serpinb1, is a vital gatekeeper for caspase-1/11 activation[42]. Serpinb1 is highly expressed on the neu-trophil surface and is critical to their survival[43]. The expression of serpinb1 was significantly downregulated after LS plus IFN-γ stimula-tion; this effect could be rescued by knocking out FcγR or upon IgG depletion (Fig. 5g, i). Strikingly, the expression of Serpinb1 in peripheral blood neutrophils isolated from SLE patients was almost completely abolished, while caspase-1 and −4 were activated (Fig. 5k, l). In addition, reduced expression of Serpinb1 activated caspases-1 and −11 in the bone marrow neutrophils isolated from the PIL mice (Fig. 5m, n). The data suggested that ICs promoted GSDMD cleavage via the FcγR/Serpinb1/caspase-1, 11-dependent pathway.

## Ox-mtDNA promotes GSDMD-N oligomerization

The most important function of GSDMD-N is the formation of octa-mer oligomer, thus inducing GSDMD pore formation on the plasma membrane[7,8]. However, the precise mechanism regulating GSDMD-N oligomerization is poorly understood. GSDMD oligomerization could be detected in the bone marrow neutrophils from PIL mice and per-ipheral blood neutrophils from SLE patients (Fig. 6a–d). Pretreatment of mouse neutrophils with the Mito-TEMPO following stimulation with LS plus IFN-γ resulted in decreased levels of GSDMD oligomer (Fig. 6e, f), however, the levels of GSDMD-N remained unaffected (Fig. 6g, h). To confirm whether mROS promoted GSDMD oligomer-ization in lupus mice, Mito-TEMPO was continuously administered prophylactically for seven weeks via subcutaneous pump to MLR/lpr lupus mice, starting at 10 weeks of age. GSDMD oligomerization was detected in neutrophils from MRL/lpr mice but not MRL/mpj mice (Fig. 6i, j). Both GSDMD oligomerization and GSDMD cleavage were reduced after Mito-TEMPO treatment (Fig. 6i–l). Because mROS can induce oxidative damage to mtDNA, we hypothesized that Ox-mtDNA could promote GSDMD oligomerization. To verify this, we used a cell-free system by incubating *Escherichia coli*-purified caspase-4 with recombinant human GSDMD; the addition of LPS can activate caspase-4 to process GSDMD into fragments[44]. GSDMD oligomeriza-tion increased significantly after Ox-mtDNA treatment, and to a lesser extend mtDNA pre-incubation (Fig. 6m, n). Ox-mtDNA-dependent GSDMD oligomerization was also in a dose-dependent manner (Fig. 6o, p). In another cell-free system of active caspase-1 and GSDMD, we also identified that GSDMD oligomer was increased after mtDNA or Ox-mtDNA treatment (Supplementary Fig. 3a, b). Ox-mtDNA has been shown to activate NLRP3 inflammasome via direct binding[45]. However, inhibition of NLRP3 in mouse neutrophils has no effect on GSDMD oligomerization (Supplementary Fig. 3c, d). In addition, pretreated LS with DNase I suppressed GSDMD oligomer-ization (Supplementary Fig. 3e, f), indicating that nuclear DNA and mtDNA in LS may also promote GSDMD oligomerization.

To further explore the role of intracellular mtDNA in GSDMD oligomerization. We pretreated mouse neutrophils with inhibitors of mitochondrial transcription (IMTs) that cause a dose-dependent decrease in the levels of mitochondrial transcripts and gradual depletion of mtDNA[46]. We also performed a combination of the inhi-bitor of VBIT-4 (outer mitochondrial membrane pore VDAC1 oligo-merization inhibitor) and CsA (binds cyclophilin D and inhibits Ca2+ regulated mPTP opening) to fully suppressed the release of mtDNA form mitochondria to the cytoplasm[47]. The results showed both IMT1 and CsA with VBIT-4 inhibits GSDMD oligomerization (Fig. 6q, r). However, the level of ROS was not suppressed by IMT1 and CsA with VBIT-4 (Supplementary Fig. 3g, h), indicating an independent role of ROS in GSDMD oligomerization in neutrophils. Thus, mtDNA was essential for GSDMD oligomerization.

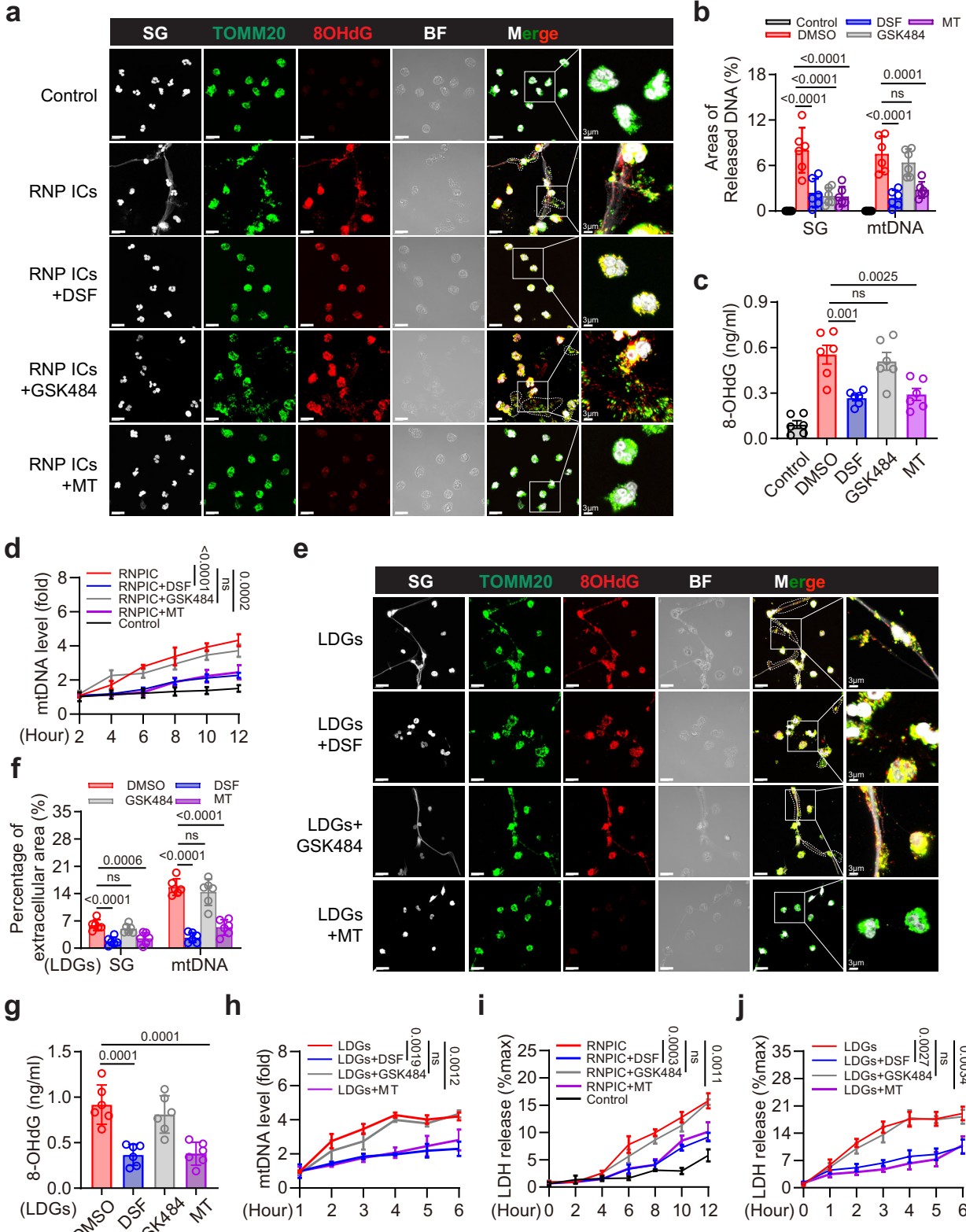

## Ox-mtDNA directly interacts with GSDMD-N

We then investigated the mechanisms underlying Ox-mtDNA-mediated GSDMD oligomerization. Co-immunoprecipitation with GSDMD could pulldown mtDNA in cell lysates stimulated with LS and IFN-γ, or classic inflammasome stimuli (LPS plus nigericin) with hydrogen peroxide, thus indicating a direct interaction between GSDMD and mtDNA (Fig. 7a). In contrast, GSDMD-pulled mtDNA was

negligibly detected in LPS plus nigericin without hydrogen peroxide, thus indicating the role of ROS in mediating the interaction between GSDMD and mtDNA. Mito-TEMPO was added and GSDMD immuno-precipitation was performed with 8OHdG and TOMM20 in murine neutrophils. Both 8OHdG and TOMM20 immunoprecipitated with GSDMD after IFN-γ plus LS treatment, and were significantly sup-pressed upon Mito-TEMPO treatment (Fig. 7b, c). We also detected a

**Fig. 4 | RNP ICs-induced release of extracellular mtDNA is significantly suppressed by mROS and GSDMD inhibition. a** Immunofluorescence staining of SG, TOMM20 and 8OHdG in peripheral blood neutrophils from HV after RNP ICs treatment. Cells were pretreated with DSF (5 μM), GSK484 (10 μM), or Mito-TEMPO (10 μM) for 2 h, followed by stimulation for 12 h with RNP ICs. BF: bright field; scale bar, 15 μm. 3 μm (enlarged). **b** Quantitative analysis of areas of extracellular SG and mtDNA in (**a**). Areas of released mtDNA including regions of extracellular TOMM20/8OHdG positive staining. **c** Quantification of 8OHdG content in culture medium after the indicated treatment. **d** Quantitative analysis of mtDNA into the supernatant from the indicated treatment groups at the indicated time points by qPCR. n = 3 HVs. **e** Immunofluorescence staining of SG, TOMM20 and 8OHdG in LDGs after 6 h. LDGs were pretreated with DSF (5 μM), GSK484 (10 μM) or Mito-TEMPO (10 μM). Scale bar, 15 μm. 3 μm (enlarged). **f** Quantitative analysis of areas of extracellular SG and mtDNA in (**e**). **g** Quantification of 8OHdG content in cultured medium of LDGs. **h** Quantification of the released mtDNA in LDGs at the indicated time points. n = 3 SLE patients. **i, j** Quantification of the release of LDH into the supernatant from the indicated groups at the indicated time points by ELISA. n = 3 HVs (**i**) and 3 SLE patients (**j**). The results are pooled from three independent experiments using cells from 6 HVs (**b, c**). Two SLE patients were used in one experiment, and plots were pooled from three independent experiments using cells from 6 SLE patients (**f, g**). Representative of three independent experiments (**a, d, e, h, i, j**). Data are presented as mean ± SD. Significance was examined by one-way AVOVA (**b, c, f, g**) or unpaired two-sided Student's *t* test (**d, h, i, j**).

direct interaction of Ox-mtDNA and GSDMD in neutrophils from SLE patients (Fig. 7d, e). GSDMD contains functionally GSDMD-N and GSDMD-C domains, which are auto-inhibitory in stabilizing full-length GSDMD[10]. To determine the association domain in GSDMD with mtDNA, HEK293T cells or *Gsdmd*[−/−] MEF cells transfected with full-length, N-terminal, or C-terminal GSDMD plasmids. Cells were treated with hydrogen peroxide to induce mitochondrial stress. Western blot showed a constitutively expression of GSDMD, GSDMD-N and GSDMD-C in HEK293T cells or *Gsdmd*[−/−] MEF cells after indicated plasmid transfection (Supplementary Fig. 4a–l). MtDNA was detected in lysates from full-length or N-terminal but not C-terminal GSDMD transfected cells (Fig. 7f, g). The results indicated a direct interaction between GSDMD-N and mtDNA. The negatively charged backbone of Ox-mtDNA can potentially interact with the positively charged residues in GSDMD-N. To identify the potential residues in GSDMD-N, we searched for evolutionarily conserved, positively charged residues in GSDMD-N by comparing the sequences of six mammalian species using the Clustal Omega and SOPMA secondary structure prediction server[48]. A cluster of four such residues occurred in a pair of predicted amphipathic α-helices (mouse Arg138, Lys146, Arg152, and Arg154) (Fig. 7h). The X-ray crystal structure of full-length GSDMD depicted four basic residues on helices α2 and α3 in mouse GSDMD-N (Fig. 7i). To directly evaluate the interaction between GSDMD and mtDNA, we measured the binding dissociation constants (Kd) of purified GSDMD protein with mtDNA or Ox-mtDNA by microscale thermophoresis (MST). mtDNA could interact with purified GSDMD protein with a Kd of 1.3 μM, whereas Ox-mtDNA showed reduced Kd indicating a stronger association (Fig. 7j, k). We further synthesized a peptide fragment of GSDMD-N containing these residues with a His-tag (QHERHLQQPEN-KILQQLRSRG). GSDMD-N could interact with Ox-mtDNA with a Kd of 239 nM; its association with mtDNA was relatively weak (Fig. 7l, m). We then mutated a cluster of these four conserved positively charged residues. Indeed, MST assay showed that both mtDNA and Ox-mtDNA had a poor fitting with mutant peptides (Fig. 7n, o). Collectively, these results showed that Ox-mtDNA could directly interact with the four positively charged residues in amphipathic α-helices of GSDMD-N.

## GSDMD deficiency results in decreased disease activity in lupus model

To directly examine the requirement for GSDMD as a mediator of SLE progression, we compared disease pathogenesis in GSDMD-knockout and their littermate (WT) mice following pristane administration. RNA-seq analysis of kidney tissues from WT and *Gsdmd*[−/−] mice revealed suppression of genes related to pro-inflammatory cytokine production or type I IFN signaling (such as TRIM5, ISG15, MX1, and IRF7) in pristane-treated *Gsdmd*[−/−] mice (Fig. 8a). The top biological GO terms enriched for genes that were differentially abundant between WT and *Gsdmd*[−/−] mice following pristane treatment were related to leukocyte activation, innate immune responses, regulation of cytokine production, and immune responses (Fig. 8b). Gene set enrichment analysis (GSEA) revealed significant enrichment of the IFN pathway and inflammatory responses upon the comparison of WT and *Gsdmd*[−/−] mice following

pristane treatment (Fig. 8c). Consistent with these findings, serum IFN-α and IL-1β levels were reduced in *Gsdmd*[−/−] mice relative to the WT mice after pristane treatment (Fig. 8d, e). GSDMD knockout rescued the severe disease-related facial damage and splenomegaly after pristane treatment (Supplementary Fig. 5a–d). The survival rate of PIL mice from 9–12 months was improved in the *Gsdmd*[−/−] mice (Supplementary Fig. 5e). Antinuclear antibody positivity is a key diagnostic criterion associated with SLE. Following pristane treatment, WT mice exhibited significantly elevated anti-ANA, anti-ssDNA, anti-RNPs and anti-Sm antibody levels (Fig. 8f, g; Supplementary Fig. 5f, g), whereas these auto-antibodies were present at significantly lower levels in the *Gsdmd*[−/−] mice. Serum level of IL-18 and mtDNA were significantly increased in PIL mice compared with control. And the increased level of IL-18 and mtDNA in PIL mice were suppressed in *Gsdmd*[−/−] mice after pristane treatment (Supplementary Fig. 5h, i). The percentages and numbers of monocytes, macrophages, and neutrophils in the spleen were decreased in the *Gsdmd*[−/−] mice after pristane treatment, whereas the numbers and relative frequencies of cDCs remained unchanged (Supplementary Fig. 6a–c). The percentages and numbers of naïve B, CD4⁺T, and effector T cells (CD44⁺CD62L⁻) in the spleen were reduced in the *Gsdmd*[−/−] mice as compared to WT mice following pristane treatment (Supplementary Fig. 6d–f). After pristane treatment for seven months, histological examination of kidney tissues revealed severe proliferative glomerulonephritis in WT mice which was rescued in the *Gsdmd*[−/−] mice (Fig. 8h). In addition, pristane-induced renal ICs deposition and proteinuria were also decreased in the *Gsdmd*[−/−] mice (Fig. 8i, j; Supplementary Fig. 5j, k). To precisely evaluate the contribution of GSDMD in neutrophils, we used a mouse strain with specifically depleted *Gsdmd* in neutrophils (Supplementary Fig. 7a)[49]. Similar to GSDMD-deficient mice, GSDMD conditional knockout in neutrophils suppressed serum IFN-α and IL-1β levels after pristane treatment (Fig. 8k, l). Serum levels of anti-ANA, anti-ssDNA, anti-RNPs, and anti-Sm (Fig. 8m, n; Supplementary Fig. 7b, c) were suppressed in the *Gsdmd*[fl/fl]S100A8-cre mice as compared to the *Gsdmd*[fl/fl] mice after pristane treatment. The level of IL-18 and mtDNA was also suppressed in *Gsdmd*[fl/fl]S100A8-cre mice as compared to the *Gsdmd*[fl/fl] mice after pristane treatment (Supplementary Fig. 7d, e). Histological examination of kidney tissues revealed severe proliferative glomerulonephritis and high kidney biopsy score in WT mice which was alleviated in the *Gsdmd*[fl/fl]S100A8-cre mice (Supplementary Fig. 7f, g). The level of proteinuria was suppressed in *Gsdmd*[fl/fl]S100A8-cre mice as compared to the *Gsdmd*[fl/fl] mice after pristane treatment (Supplementary Fig. 7h). In addition, renal ICs deposition and proteinuria also decreased in the *Gsdmd*[fl/fl]S100A8-cre mice as compared to the *Gsdmd*[fl/fl] mice after pristane treatment (Supplementary Fig. 7i–k).

## Pharmacological inhibition of GSDMD attenuates lupus disease activity

As GSDMD-deficient mice exhibited impaired SLE pathogenesis, we sought to test whether the pharmacological inhibition of GSDMD was sufficient to achieve a similar degree of disease remission in the MRL/lpr mice. Thus, we orally administered the GSDMD inhibitor, DSF, to these mice. Disease-related increases in splenic and lymph node

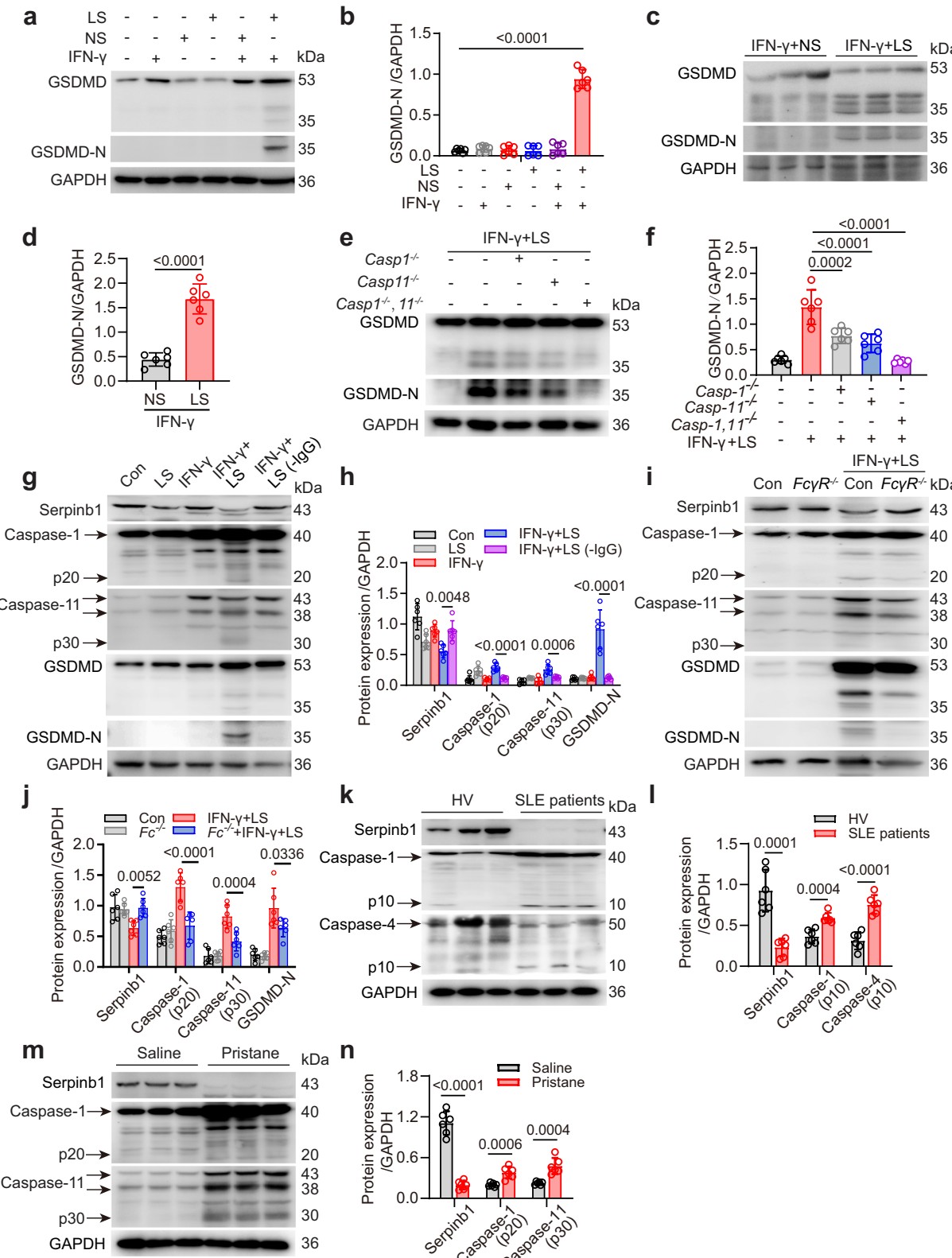

weight in MRL/lpr mice were significantly suppressed after DSF treatment (Fig. 9a–d). Reduced levels of anti-dsDNA, anti-RNPs, and anti-Sm (Fig. 9e–g) autoantibodies in MRL/lpr mice following DSF treatment were observed. Serum levels of IL-1β, IL-18 and mtDNA were also reduced in the MRL/lpr mice after DSF treatment (Fig. 9h–j). Kidney histopathological analyses revealed severe glomerular, interstitial, and vascular lesions in the vehicle-treated MRL/lpr mice consistent with

glomerulosclerosis, crescent formation, increased mesangial matrix, tubular atrophy, and diffuse perivascular and interstitial mononuclear cell infiltration, all of which were alleviated following DSF treatment (Fig. 9k, l). Kidney function measured using the urine albumin-to-creatinine ratio (UACR) improved after DSF treatment (Fig. 9m). DSF treatment also decreased renal deposition of ICs (IgG and Complement C3) (Fig. 9n–p). These data indicated that DSF-mediated GSDMD

**Fig. 5 | Immune complexes mediate the activation of caspase-1 and caspase-11 by downregulating Serpinb1. a** Immunoblotting of GSDMD and GSDMD-N in BM neutrophils after treatment with IFN-γ, NS from WT mice, LS from lupus mice, or IFN-γ + LS for 12 h. **b** Quantitative analysis of GSDMD-N/GAPDH. **c** Immunoblotting of GSDMD and GSDMD-N in peripheral blood neutrophils of HV. Cells were pretreated with IFN-γ. Then the cells were added with NS form HV or LS from SLE patients for 12 h. **d** Quantitative analysis of GSDMD-N/GAPDH levels. **e** Immunoblotting for GSDMD and GSDMD-N in BM neutrophils from *Caspase-1^-/-*, *Caspase-11^-/-* and *Caspase-1^-/-Caspase-11^-/-* mice treated with IFN-γ + LS. **f** Quantitative analysis of GSDMD-N/GAPDH levels. **g** Immunoblotting of Serpinb1, Caspase-1, Caspase-11, and GSDMD levels in BM neutrophils treated with IFN-γ + LS, or IgG deleted LS for 12 h. **h** Quantitative analysis of Serpinb1, Caspase-1, Caspase-11, and GSDMD-N levels. **i** Immunoblotting analysis of Serpinb1, Caspase-1, Caspase-11,

and GSDMD levels in WT or *FcγR^-/-* BM neutrophils after IFN-γ + LS treatment for 12 h. **j** Quantitative analysis of Serpinb1, Caspase-1, and Caspase-11 levels. **k** Immunoblotting analysis of Serpinb1, Caspase-1, and Caspase-4 levels in neutrophils isolated from the peripheral blood of HV and SLE patients. **l** Quantitative analysis of Serpinb1, Caspase-1, and Caspase-4 levels. **m** Immunoblotting analysis of Serpinb1, Caspase-1, and Caspase-11 levels in the BM neutrophils from saline or PIL mice. **n** Quantitative analysis of Serpinb1, Caspase-1, and Caspase-11 levels. n = 6 mice (**b, f, h, j, n**) or 6 donors (**d, l**). The immunoblotting samples shown are from the same experiment. Two blots were processed in parallel (**d, l, n**). Three blots were processed in parallel (**b, f, h, j**). Data are shown as mean ± SD. Significance was examined by one-way ANOVA (**b, f, h, j**), or unpaired two-sided Student's *t* test (**d, l, n**).

inhibition was sufficient to delay disease progression in the MRL/lpr mice.

Collectively, ICs and IFN-γ in the serum of SLE promote the activation of GSDMD through serpinb1/caspase-1/11 pathway. Meanwhile, ICs also promotes mitochondrial stress, which facilitates the release of Ox-mtDNA into the cytosol. Cytosolic Ox-mtDNA directly binds with GSDMD-N to promote GSDMD-N oligomerization and GSDMD pore formation. At last, the subsequent extracellular release of NETs and mtDNA promote SLE pathogenesis (Fig. 9q).

## Discussion

There has been an explosion of interest in understanding the role of cell death and DAMP signaling pathways in shaping inflammatory and immune responses[50]. Herein, we performed intravital imaging and molecular biology to examine the relationship between pyroptosis-driven protein, GSDMD, and interferogenic mtDNA in neutrophils in the context of SLE. We confirmed that not only does GSDMD promote extracellular DNA release but the oxidation of mtDNA is essential for GSDMD-N oligomerization and subsequent cell death. Thus, the interaction between Ox-mtDNA and GSDMD results in a positive feedback loop, thereby ensuring that extracellular DNA release and pro-inflammatory programmed cell death in neutrophils in SLE.

MtDNA is present at high cellular copy number and more prone to damage than nuclear DNA[51]. Previous studies of mtDNA in autoimmune disease mainly focused on the mechanism that regulates the release of mtDNA from mitochondria. Benjamin Kile discovered mtDNA escaped from mitochondria through BAK/BAX membrane pores, and confirmed that mtDNA released from mitochondria is an important factor in SLE[50]. Subsequently, there are studies showed that voltage-dependent anion channel (VDAC) in the mitochondrial membrane facilitates cytosolic mtDNA release in SLE, and TAR DNA-binding protein of 43 kDa (TDP-43) in cytoplasm promotes cytosolic mtDNA release in familial amyotrophic lateral sclerosis[52,53]. Our study showed cytosolic mtDNA directly interacts with GSDMD-N. However, the precise mechanism governing the release of mtDNA from mitochondria (BAK/BAX, VDAC pore or TDP-43) after ICs treatment needs further investigation.

Cytosolic mtDNA is best known for activating cGAS/STING and promoting type I IFN production[54]. However, neutrophils do not express a relative high level of type I IFN. Our results indicated a novel role of cytosolic mtDNA that interacts with GSDMD-N and promotes GSDMD-dependent NETs. A number of other studies showed that mtDNA also activates Toll-like receptor (TLR) 9[55], NLRP3 inflammasome and other cytoplasmic nucleic acid sensors (e.g., Zbp1)[45,56]. In addition, previous studies identified that neutrophils in SLE extrude Ox-mtDNA outside the cell[21,22]. Our results also showed the release of mtDNA in cultured medium was significantly suppressed by GSDMD deficiency. In consistent with a previous study in macrophage, GSDMD was identified to promote a fast mitochondrial collapse, cytosolic mtDNA accumulation, and mtDNA release from cells during pyroptosis and apoptosis. They further revealed that GSDMD pores were not big enough to allow mtDNA to be released from the cell, but are big

enough to induce its release from the mitochondrial matrix[57]. Our results showed that GSDMD knockout significantly suppressed extracellular mtDNA release after LS plus IFN-γ treatment. The discrepancy might be due to the different cell types and stimulations. It was identified that cell-free supernatants from healthy neutrophil cultures contain mtDNA in the absence of activation, while monocytes extrude negligible amounts of mtDNA[21]. Furthermore, a recently study proved cytosolic Ox-mtDNA are not intact circular DNA, but 500–650 bp fragments[47]. So GSDMD is sufficient to promote extracellular mtDNA release in neutrophils after LS plus IFN-γ treatment.

As the key executioner of pyroptosis, GSDMD oligomerizes and forms pore-like structures on the plasma membrane upon inflammatory stimulation[7,9]. Herein, we identified mROS as a new regulator of GSDMD, consistent with the findings of a previous report, whereby the regulator-Rag-mTORC1 was shown to promote GSDMD oligomerization through ROS production[15]. However, mechanistic insights into the effects of mROS on GSDMD oligomerization are unknown. Herein, we found that cytosolic Ox-mtDNA released from mitochondria could directly interact with GSDMD and promotes GSDMD-N aggregation during pore formation. Thus, as a decisive factor, Ox-mtDNA could stabilize the GSDMD-N oligomer by interacting with its positively charged residues. Furthermore, the clinical relevancy of our findings is strongly supported by the level of cleaved GSDMD-N and its positive correlation with serum levels of NETs-associated DNA and mtDNA in peripheral blood neutrophils from SLE patients as well as the reduction of lupus severity in MRL/lpr mice after systemic inhibition with GSDMD inhibitor.

The role of GSDMD has not been well characterized in the context of SLE. A recent study using imiquimod (IMQ)-induced SLE model and pristane-induced acute lung injury suggests that GSDMD negatively regulates autoantigen generation and immune dysregulation[58]. However, GSDMD is widely expressed in many different cell types and thus the opposite function of GSDMD in these models may indicate its roles in other cell types. Nevertheless, in our study, we have generated both conventional GSDMD knocked out and neutrophil-specific GSDMD knocked out mice to precisely dissect the contribution of GSDMD in neutrophils in pristane-induced SLE. The pristane-induced-lupus model is the most widely used murine model of induced-lupus disease which is highly dependent on the overproduction of type I IFNs, similar to that in over half of patients with lupus[59,60]. Both genetic and environmental factors contribute to lupus in humans, and therefore, PIL mouse model represents an environmental factor inducing lupus-like disease in a strain that is not genetically prone to autoimmune diseases.

GSDMD can bind to cardiolipin, a mitochondrial membrane component. Activated GSDMD can form mitochondrial pores and mediate the release of mtDNA into the cytosol of endothelial cells[61]. Our data indicated that the release of Ox-mtDNA from neutrophils after RNP ICs treatment or LDGs could be suppressed by the inhibition of GSDMD. We also demonstrated a direct interaction between GSDMD and mtDNA through a series of

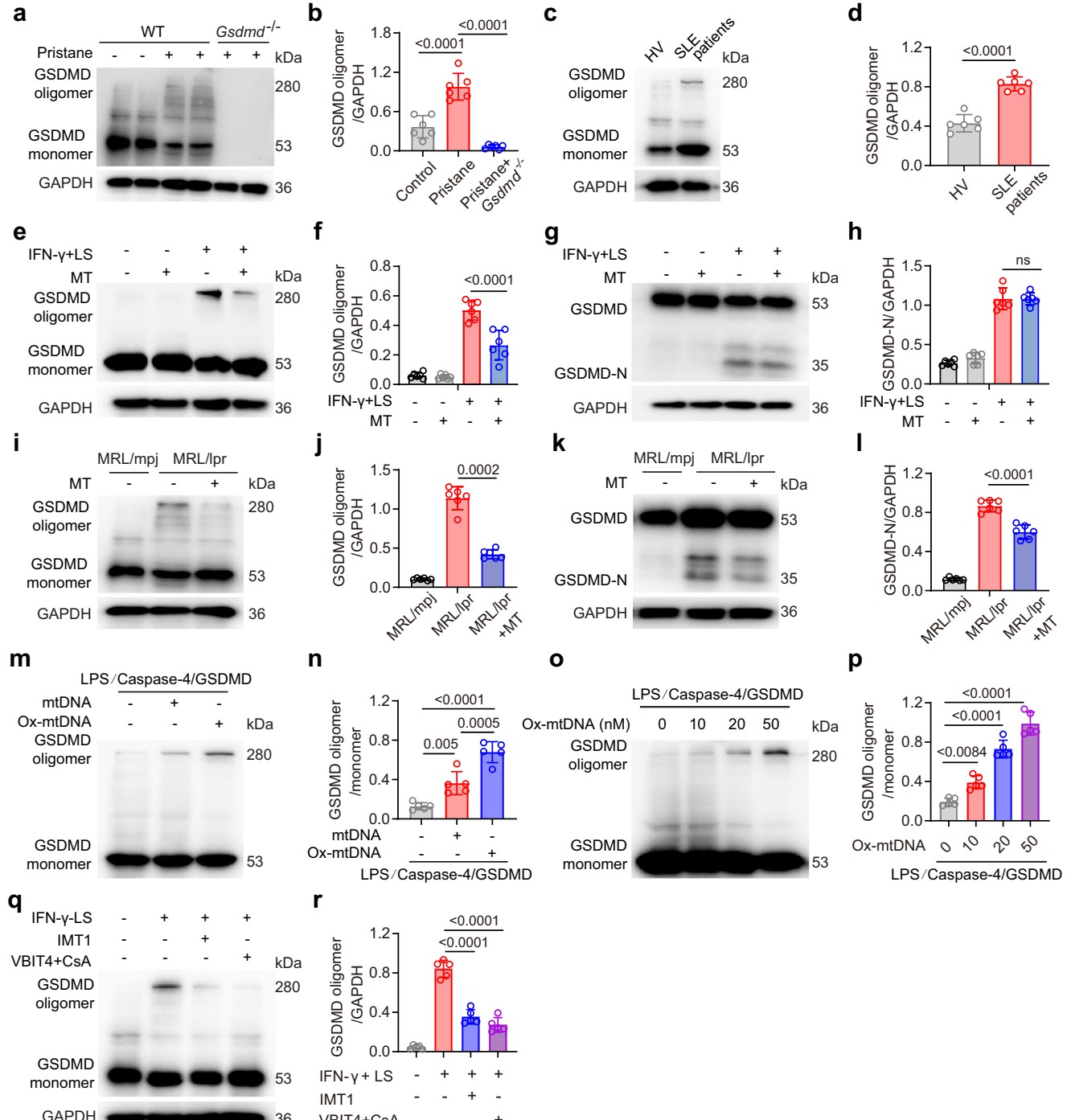

**Fig. 6 | Oxidized mtDNA promotes GSDMD oligomerization. a** Non-reducing immunoblotting of GSDMD oligomer in BM neutrophils isolated from WT, PIL mice and *Gsdmd⁻/⁻* mice after pristane treatment. **b** Quantitative analysis of GSDMD oligomer/GAPDH. **c** Non-reducing immunoblotting of GSDMD in neutrophils from HV and SLE patients. **d** Quantitative analysis of GSDMD oligomer/GAPDH. *n* = 6 HVs or 6 SLE patients. **e** Non-reducing immunoblotting of GSDMD oligomer in murine BM neutrophils. Cells were treated with IFN-γ plus LS, or pretreated with Mito-TEMPO. **f** Quantitative analysis of GSDMD oligomer/GAPDH. **g** Immunoblotting of GSDMD in murine BM neutrophils. Cells were treated with IFN-γ plus LS, or pretreated with Mito-TEMPO. **h** Quantitative analysis of GSDMD-N/GAPDH. **i** Non-reducing immunoblotting of GSDMD oligomer in BM neutrophils isolated from MRL/mpj, MRL/lpr or MRL/lpr after Mito-TEMPO treatment. **j** Quantitative analysis of GSDMD oligomer/GAPDH. **k** Immunoblotting of GSDMD in BM neutrophils isolated from MRL/mpj, MRL/lpr and MRL/lpr after Mito-TEMPO treatment.

**l** Quantitative analysis of GSDMD-N/GAPDH. **m** Non-reducing immunoblotting of GSDMD oligomer. Purified GSDMD protein was incubated with purified caspase-4 protein and LPS in vitro, then added with human mtDNA or Ox-mtDNA (20 nM). **n** Quantitative analysis of GSDMD oligomer/GSDMD monomer. **o** Non-reducing immunoblotting of GSDMD oligomer. The LPS/caspase-4/GSDMD system was added with Ox-mtDNA at 0, 10, 20 and 50 nM. **p** Quantitative analysis of GSDMD oligomer/GSDMD monomer. **q** Non-reducing immunoblotting of GSDMD oligomer. Murine BM neutrophils were pretreated with IMT1 or CsA plus VBTI4 and then treated with IFN-γ plus LS. **r** Quantitative analysis of GSDMD oligomer/GAPDH. n = 5 (**r**) or 6 mice (**b**, **d**, **f**, **j**, **l**). Plots were pooled from five independent experiments (n, p). The immunoblotting samples shown are from the same experiment. Three blots were processed in parallel (**b**, **d**, **f**, **h**, **j**, **i**, **r**). Data are presented as mean ± SD. Significance was examined with one-way ANOVA (**b**, **f**, **h**, **j**, **l**, **n**, **p**, **r**) or unpaired two-sided Student's *t* test (**d**).

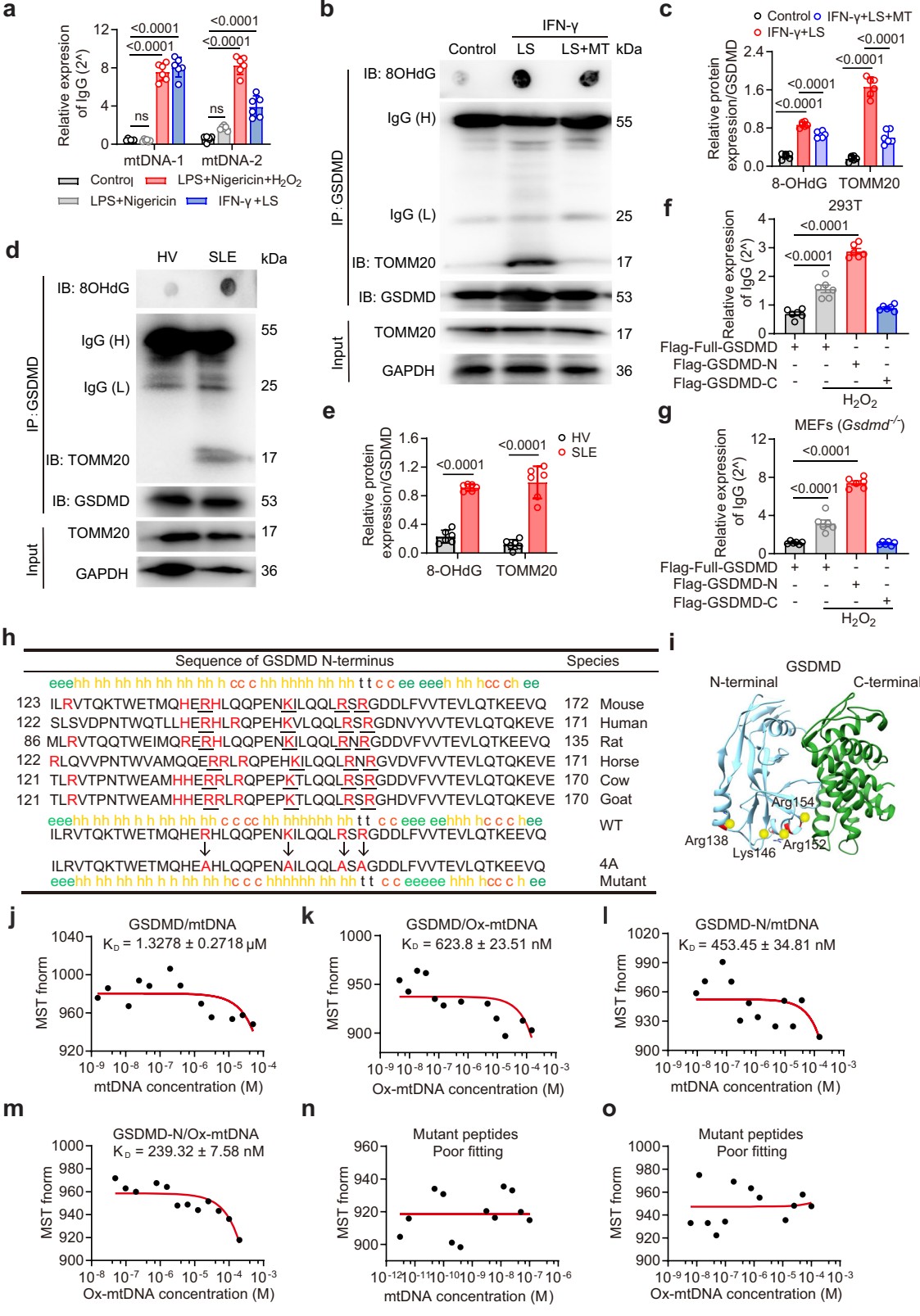

immunoprecipitation experiments and in cell-free systems. Interestingly, mtDNA can also interact with VDAC1 in the outer mitochondrial membrane. This interaction stabilizes VDAC oligomer state and further increases mtDNA release[52]. Therefore, we proposed that the interaction of mtDNA and GSDMD may help stabilize the GSDMD pore, thus creating a feedforward cycle to promote GSDMD oligomerization and increase mtDNA release.

After activation, caspase-1/11 can readily cleave GSDMD at the aspartate residue within the linking loop region. In neutrophils, such cleavage can also be mediated by other serine proteases, including elastase and cathepsin G[3,62]. GSDMD cleavage following serum treatment was partially suppressed upon knocking out caspase-1 or caspase-11, while it was fully ablated in caspase-1/11 double deficient mice. Why other proteases do not play a significant role in mediating GSDMD

**Fig. 7 | MtDNA directly interacts with GSDMD-N. a** qPCR analysis of mtDNA following GSDMD pulldown under the indicated treatment. BM neutrophils were treated with LPS + Nigericin + $H_2O_2$, LPS + Nigericin, or IFN-γ + LS. $n = 6$ mice. **b** Immunoblotting of TOMM20 and GSDMD. BM neutrophils were treated with IFN-γ + LS for 12 h, or pretreated with Mito-TEMPO. Lysates were co-immunoprecipitated with anti-GSDMD, and co-immunoprecipitates were then spotted on a nitrocellulose membrane, UV crosslinked, and probed with antibodies specific for 8OHdG, or were separated via SDS-PAGE for GSDMD and TOMM20. **c** Quantitative analysis of 8OHdG and TOMM20. $n = 5$ mice. The samples shown are from the same experiment. Three blots were processed in parallel. **d** Immunoblotting of TOMM20 and 8OHdG in peripheral blood neutrophils from HV or SLE patients ($n = 6$). **e** Quantitative analysis of 8OHdG and TOMM20. The samples shown are from the same experiment. Three blots were processed in parallel. Relative mtDNA enrichment was assessed via qPCR in indicated cells. 293T

(**f**) and *Gsdmd*$^{-/-}$ MEF cells (**g**) were transfected with Flag-full-GSDMD, Flag-GSDMD-N or GSDMD-C, then treated with $H_2O_2$ (100 μM) for 4 h. $n = 6$ samples pooled from 6 independent experiments. **h** The cluster of four evolutionarily conserved, positively charged amino acids (red and underlined) in GSDMD-N were mutated to Ala. **i** X-ray crystal structure of the murine GSDMD (PDB: 6N9N). Dissociation constants ($K_D$) of human GSDMD with human mtDNA (**j**) and Ox-mtDNA (**k**). $K_D$ of peptides from GSDMD-N with human mtDNA (**l**) and Ox-mtDNA (**m**). **n, o** Measurements of $K_D$ of mutant peptides from GSDMD-N with mtDNA (**n**) and Ox-mtDNA (**o**). The $K_D$ was derived from the binding response as a function of the His-tagged GSDMD or His-tagged peptides. Errors in $K_D$ represent fitting errors. Representative of three independent experiments (**b, d, j–o**). Data are presented as means ± SD. Significance was examined with one-way ANOVA (**a, c, f, g**) or unpaired two-sided Student's $t$ test (**e**).

cleavage in the context of SLE remains to be clarified. A previous study through proteomic analysis showed that NETs induced in response to different stimuli exhibit distinct patterns of protein composition[63,64]. Thus, it is possible that the upstream factors related to GSDMD activation are context-dependent and may be linked to different outcomes. Both caspase-1 and −11 are required for GSDMD activation upon exposure to lupus serum. However, several well-known upstream activators of caspases, such as danger signals and bacteria toxins, are not likely to be present at high concentrations in lupus serum. In a recent study, Jung et al. report that Serpinb1, a protease inhibitor expressed at high levels in neutrophils, serves as a key gatekeeper for the activation of caspase-1 and -11[42]. Our data clearly indicated that ICs treatment could suppress the safeguarding function of Serpinb1 in vitro. Unexpectedly, the expression of Serpinb1 in the neutrophils of SLE patients and PIL mice were extremely low or absent, suggesting that serpinb1-dependent caspase-1/11 activation may be a constantly ongoing process in the context of SLE.

The FDA-approved drug disulfiram (DSF), which is used to treat alcohol addiction, was recently shown to inhibit GSDMD pore formation[65]. We found that DSF was able to inhibit neutrophil extracellular DNA release and to suppress Ox-mtDNA release from LDGs. DSF administration to lupus-prone mice ameliorated lupus symptoms. However, it is worth to mention that DSF is known to modify cysteine residues through formation of a disulfide bridge and thus may have many other target proteins[66,67]. The effect of DSF treatment in lupus model may not be solely attributed to GSDMD inhibition. Nevertheless, given the significant unmet clinical needs of SLE patients and particularly of individuals suffering from lupus nephritis, the therapeutic potential for the repurposing of this drug as a means of treating SLE warrants further study.

## Methods
### All animal experiments were approved by Shanghai Jiao Tong University School of Medicine
Institutional Animal Care and Use Committees (IACUC) and performed according to local guidelines for the use and care of laboratory animals as provided by Shanghai Jiao Tong University School of Medicine IACUC (protocol No. A-2022-042). Studies enrolling human neutrophils were approved by the Research Ethics Committee of Shanghai Jiao Tong University College of Basic Medical Sciences (protocol No. 2022-23). Human renal biopsies were obtained under a protocol approved by the Ethics Committee of Shanghai Jiao Tong University Affiliated Shanghai Tong Ren hospital (protocol No. 2017-004-01).

### Animals
Female, 4 weeks C57BL/6 mice were purchased from SLRC laboratory animal (Shanghai, China). *Gsdmd*$^{-/-}$ mice were obtained from the model animal research center of Nanjing University (T010437). *Casp11*$^{-/-}$ mice were obtained from Jackson Laboratory (#024698). *Casp1*$^{-/-}$*Casp11*$^{-/-}$ mice were obtained from Jackson Laboratory

(#016621). *Casp1*$^{-/-}$ mice were obtained from Cyagen biotechnology (S-KO-01331). MRL/mpj (000486), MRL/lpr (000485) mice were purchased from SLRC laboratory animal (Shanghai, China). *FcγR*$^{-/-}$ mice (Jackson Laboratory #002847) were kindly provided by Dr. Fu-bin Li. *Ms4a3tdTomato* mice were kindly provided by Dr. Florent Ginhoux[31]. *Gsdmd*$^{flox/flox}$ (S-CKO-14431) was obtained from Cyagen biotechnology. *S100A8-cre* mice (#021614) was obtained from The Jackson Laboratory. Animals were maintained under specific pathogen-free conditions. Experimental and control animals were co-housed with no more than five animals per cage. All mice were housed under specific pathogen-free conditions and kept in a 12:12-h light–dark cycle with controlled humidity (60–80%) and temperature (22 ± 1 °C). All experiments were performed on sex-and 4- to 12-week-old age-matched animals. After treatment with pristane for 7 months, mice were anesthetized with isoflurane inhalation for body wight measurement and blood collection, and then euthanized by avertin (100 mg/kg) and rapid cervical dislocation for kidney and spleen collection.

### Clinical patient information
Serum samples were collected from 34 patients with ANA-positive SLE who met the 2017 EULAR/ACR SLE. The control group consisted of 10 healthy individuals who consented to participate in our project. Informed consent was obtained from all participants. Additional clinical information about the participants is listed in Supplementary Table 1. For human renal biopsy studies, written informed consent from participants and parents or legal guardians where applicable was received under protocols approved by the ethics committee of Shanghai Jiao Tong University Affiliated Shanghai Tong Ren hospital. Patients who underwent a renal biopsy and were diagnosed with lupus nephritis without any other type of renal injury were enrolled in the study.

### Antibodies and reagents
Detailed information of antibodies used for western blot, flow cytometry and Immunofluorescence staining are listed in Supplementary Table 2. Mito-TEMPO (HY-112879), MIT1 (HY-134539), VBIT-4 (HY-129122), MCC950 (Hy-12815) and GSK484 (HY-100514) were from MedChemExpress (NJ, USA). GSDMD with N-terminal Flag tagged plasmid (MG5A2835-NF) and C-terminal Flag tagged plasmid (MG5A2835-CF) were obtained from SinoBiological (Beijing, China). DSF was obtained from Sigma-Aldrich (PHR1690, St. Louis, MO, USA). Purified His-tagged murine GSDMD, human GSDMD and human caspase-4 protein were obtained from Wuhan Chemstan Biotechnology (Wuhan, China). Recombinant human caspase-1 protein was obtained from Abcam (ab39901, Cambridge, MA, USA). The amino acid sequence of peptide from mouse GSDMD-N terminal was HHHHHHQHERHLQQPENKILQQLRSRG. The mutant sequence was HHHHHHQHERHLQQPENAILQQLASAG. The peptide was synthesized by GL biochem (Shanghai, China).

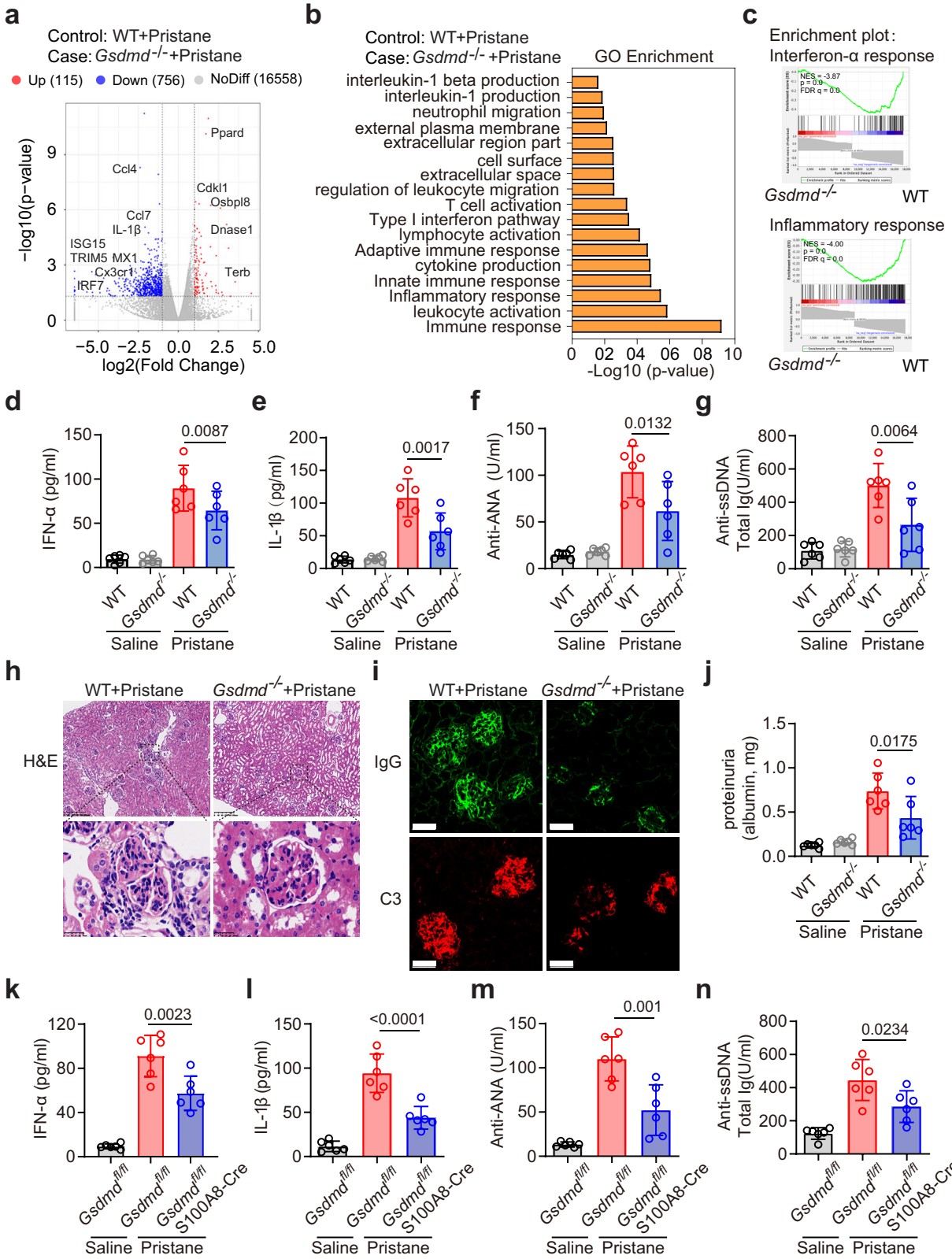

**Nature Communications** | (2023)14:872

## Lupus mouse model

Female mice (4 week) were intraperitoneally injected once with 0.5 ml of pristane oil (2,6,10,14-tetramethylpentadecane, Sigma-Aldrich). Control mice were treated with 0.5 ml of saline. The mice were euthanized by raising $CO_2$ concentrations 31 weeks after injection. Mice were anesthetized with isoflurane inhalation for serum collection. Both kidneys were excised, one for histology analysis and the other for immunofluorescent staining. Kidneys and spleens were isolated and weighed. Six-week-old female MRL/*lpr* mice were treated with DSF (50 mg/kg) through the oral administration every 2 days for 3 months.

## Isolation and stimulation of human peripheral neutrophils

Heparinized blood from SLE patients and healthy controls was isolated by density gradient centrifugation on Polymorphprep (AS1114683,

**Fig. 8 | GSDMD deficiency significantly reduced disease activity in PIL mice.** **a** Volcano plots were prepared demonstrating the distributions of P-values [−log10 (*P* value)] and fold change values [log2 (fold change)]. Down- and upregulated genes are shown in blue and red respectively. **b** Differential gene expression in kidneys from PIL mice relative to pristane-treated *Gsdmd*[−/−] mice. GO enrichment analysis demonstrating the biological processes most significantly enriched in the kidneys of WT and *Gsdmd*[−/−] mice following pristane treatment. **c** GSEA approaches identifying genes associated with IFN-α response and inflammatory response between WT and *Gsdmd*[−/−] mice following pristane treatment. ELISA analysis of serum IFN-α (**d**), IL-1β (**e**), anti-ANA (**f**) and anti-ssDNA (**g**) levels from the indicated groups. **h** H&E staining of glomeruli from WT and *Gsdmd*[−/−] mice following pristane treatment. Scale bar, 200 μm, 25 μm (enlarged). **i** Immunofluorescence of IgG (green) and complement C3 (red) in kidney section from WT and *Gsdmd*[−/−] mice after pristane treatment. Scale bar, 25 μm. **j** ELISA analysis of urine albumin levels in WT and *Gsdmd*[−/−] mice after pristane treatment. ELISA analysis of IFN-α (**k**), IL-1β (**l**), anti-ANA (**m**) and anti-ssDNA (**n**) antibodies in serum from *Gsdmd*[fl/fl] or *Gsdmd*[fl/fl]*S100A8-Cre* mice after pristane treatment for 7 months. NES normalized enrichment score, FDR false discovery rate. *n* = 6 mice (**d**–**g**, **j**–**n**) and data are representative of 2 independent experiments (**d**–**n**). Data are shown as means ± SD. Significance was examined with one-way ANOVA (**d**–**g**, **j**–**n**).

Axis-Shield, Dundee, UK) for 40 min at 400 × *g* without gracing. The neutrophils/red blood cells (RBC) were suspended in RBC lysis buffer (Servicebio) and neutrophils were washed in sterile PBS and suspended. The low-density granulocytes (LDG) were purified from peripheral blood mononuclear cells (PBMCs) isolated from fresh whole healthy and lupus donor blood using CPT-heparin tubes as previous described[68]. The EasySep™ Human Neutrophil Negative Selection (STEMCELL) Kit was used for isolation of LDG and NDG. LDG isolation purity ranged from 95–99% using this method.

### Isolation and stimulation of mouse bone marrow neutrophils
Femurs and tibias were collected and bone marrow was flushed into a 50 ml conical tube with HBSS. Following ACK lysis, neutrophils were purified through a 62.5% Percoll gradient. The purity of isolated cells has reached over 90% as determined by flow cytometry. Neutrophils were treated with 10% serum from mice treated with pristane after 7 months, and IgG-depleted serum with or without the presence of IFN-γ (100 ng/ml).

### RNP ICs isolation
RNP ICs isolation was performed as previous described[22]. Human IgG was purified through HiTrap Protein A HP (cytiva) from three individuals with SLE selected for the presence of a higher titer of anti-RNP, anti-SmRNP and anti-dsDNA antibodies. The IgG was then mixed with SmRNP (Arotec, New Zealand) and used at final concentration of 10 μg/ml.

### mtDNA pulldown
mtDNA pulldown was performed according to the previous report[69]. A total of $1 × 10^7$ bone marrow neutrophils was treated with different stimulators. Cell pellets were resuspended in polysome lysis buffer (10 mM HEPES pH 7.0, 100 mM KCl, 5 mM $MgCl_2$, 25 mM EDTA, 0.5% NP40, complete protease inhibitors) and then incubated on ice for 10 min before frozen at −80 °C for 1 h. Cell lysates were clarified by centrifugation at 13,000 r.p.m. for 15 min at 4 °C. Protein G agarose beads (Santa Cruz) were washed with NT2 buffer (50 mM Tris−HCl, pH 7.4, 150 mM NaCl, 1 mM $MgCl_2$, 0.05% NP40) for 30 min at 4 °C. The washed beads were conjugated with anti-GSDMD or IgG antibody (Sigma) at 4 °C overnight. The conjugated beads were then washed and incubated with cell lysate at 4 °C for at least 4 h. Protein G agarose beads from each condition were treated with protease K for 30 min at 55 °C and RNA or DNA was extracted using either acidic or basic phenol-cholochrome, respectively. RNA or DNA was precipitated by centrifugation for 30 min at 4 °C, washed with 80% ethanol and air-dried before resuspension in pure $H_2O$ for further use.

### Renal intravital multiphoton microscopy
Intravital imaging of kidney was performed as described previously[70]. Mice were anesthetized by an initial intraperitoneal injection of avertin (100 mg/kg). The tail vein was catheterized to allow delivery of fluorescent probes and to maintain anesthesia as required. A heating pad was used to maintain the temperature of the mice at 37 °C. The left kidney was exteriorized through a dorsal incision and the kidney was immobilized in a heated well incorporated into a custom-built stage.

The exposed kidney was bathed in normal saline and cover slipped. Glomeruli were observed using multiphoton microscope with two tunable lasers (FVMPE-RS-TWIN, Olympus). Experiments in C57BL/6 mice were performed at 1200 nm excitation, 850 nm excitation. In most experiments, images were taken every 30 s by collecting z-stacks of approximately 20 μm, with 5 μm step size. Emitted fluorescence was detected by non-descanned detectors with 495–540 nm, 575–645 nm and 660–750 nm emission filters. Pre-defined settings for laser power and detector gain were used for all experiments. For visualization of the vasculature, Qtracker® 655 (Invitrogen) was used at a concentration of 10 μM. To label DNA, Sytox green (Invitrogen) was injected through the tail vein at a concentration of 0.5 μM.

### Quantitative RT-PCR
Total RNA was extracted by using TRIzol reagent and subjected to cDNA synthesis. Quantitative RT-PCR was performed using SYBR Green Supermix (Vanzyme). The relative expression levels of mRNA were normalized against GAPDH. The sequences of qPCR primers were listed in Supplementary Table 3.

### RNA-sequencing analysis
Kidneys from different groups of mice were used for total RNA isolation with TRIzol (Invitrogen). RNA-Sequencing services were provided by Personal Biotechnology Co., Ltd. Shanghai, China using Illumina Novaseq6000 (150 bp paired end reads). The raw reads were aligned to the mouse reference genome (version mm10) by using Hisat2 RNASeq alignment software. The mapping rate was 96% in average across all the samples in the dataset. HTSeq was used to quantify the gene expression counts from TopHat2 alignment files. Using R package DESeq2, differential expression analysis was performed on the count data. *p* values obtained from multiple tests were adjusted by using Benjamini-Hochberg correction.

### Western blot
Purified bone marrow or peripheral blood neutrophils were lysed in cell lysis buffer (50 mM Tris−HCL (pH8.0), 2% SDS, 10% glycine) supplemented with proteinase inhibitor at a dilution of 1:25. Protein concentrations were determined by a BCA protein assay kit. Equal amount (30 μg per sample) of total protein was loaded and separated by SDS-PAGE and then electrophoretically transblotted onto a PVDF membrane. For detection of GSDMD oligomers induced as shown, proteins were separated in non-reducing SDS-PAGE gels. The membranes were blocked and incubated with primary antibody at 4 °C overnight. With washed in TBST, membranes were incubated with specific anti−rabbit or anti-mouse antibodies for 2 h at room temperature. The signals were then imaged with the ImageQuant LAS4000 mini system (GE healthcare).

### Immunoprecipitation (IP) and co-IP
Bone marrow neutrophils after indicated treatment or peripheral blood neutrophils from HV or SLE patients were gently lysed in ice-cold NP40 buffer (Beyotime, P0013F). Cell lysate (500 μg proteins) was firstly incubated with 1 μg IgG antibody and 8 μL Protein G beads for 1 h at 4 °C. After centrifuge 8000 g for 1 min, magnetic rack was

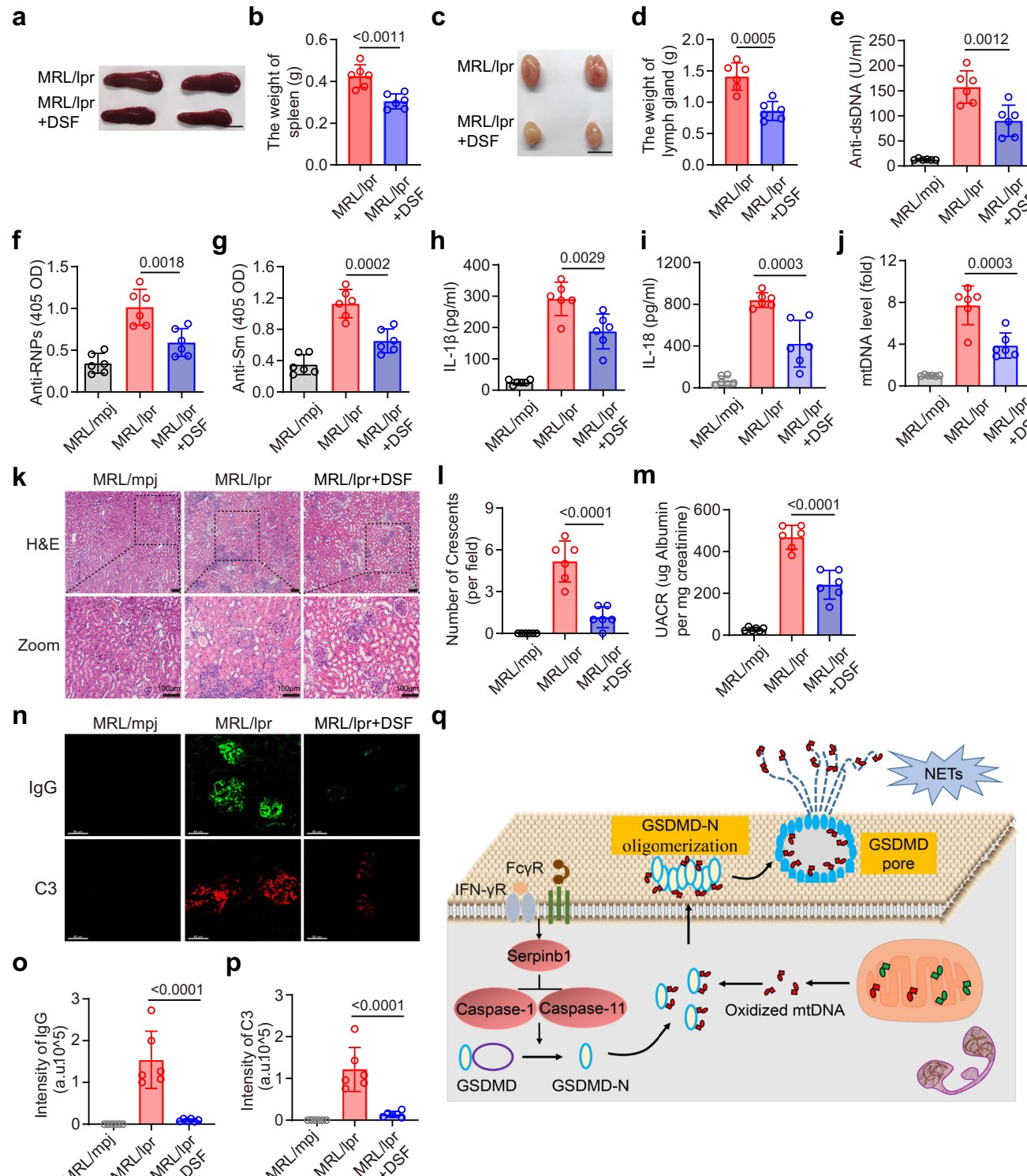

**Fig. 9 | GSDMD inhibition alleviates disease activity in MRL/lpr mice. a** Images of spleen from MRL/lpr mice or MRL/lpr with DSF treatment. Scale bar, 1 cm. **b** Quantitative analysis of spleen weight. **c** Images of lymph gland from MRL/lpr mice or MRL/lpr with DSF treatment. Scale bar, 1 cm. **d** Quantitative analysis of the weight of lymph gland (4 lymph glands). ELISA analysis of the levels of anti-dsDNA (**e**), anti-RNPs (**f**), anti-Sm (**g**), IL-1β (**h**) and IL-18 (**i**) in serum samples from the indicated groups. **j** qPCR analysis of D-loop in serum from indicated groups. **k** H&E staining of kidney section from indicated groups; scale bar = 100 μm. **l** Quantitative analysis of numbers of glomerular crescents from indicated groups. **m** ELISA of urine albumin-to-creatine ratio in MRL/mpj, MRL/lpr, and MRL/lpr mice treated with DSF. **n** Renal section from the indicated treatment groups were stained with or

IgG (green) and C3 (red). Scale bar, 50 μm. **o, p** Quantitative analysis of the intensity of IgG and C3 per FOV. **q** Working model. ICs and IFN-γ in the serum of SLE patients promote the activation of GSDMD through serpinb1/caspase-1/11 pathway. Moreover, ICs also promote mitochondrial stress, which facilitates the release of Ox-mtDNA into the cytosol. Cytosolic Ox-mtDNA directly binds with GSDMD-N to promote GSDMD-N oligomerization and GSDMD pore formation. The subsequent extracellular NETs and mtDNA promote SLE pathogenesis. *n* = 6 mice, 2 independent experiments (**b, d–j, l, m, o, p**). Data are shown as means ± SD. Significance was examined with unpaired two-sided Student's *t* test (**b, d**) one-way ANOVA (**e–j, l, m, o, p**).

used and supernatant was collected to a new tube. The supernatant was then incubated overnight with anti-GSDMD antibody (2 μl) at 4 °C. Subsequently, 20 μl of protein A/G agarose (Santa Cruz Biotechnology, Inc) were added for additional 2 h. The beads were then washed two times with NETN300 (high salt, 60 mM Tris-HCl, 3 mM EDTA, 1.5% NP40, 300 mM NaCl), subsequently two times with NETN100 (low salt, 20 mM Tris-HCl, 1 mM EDTA, 0.5% NP40, 100 mM NaCl). The associated complexes/proteins were released from the immunocomplexes by incubation for 5 min at 100 °C with 2× loading buffer. The dissociated complexes were then collected by centrifugation and subjected to SDS-PAGE for Western blotting analysis.

## Dot plot assay

Cells were cross-linked by UV irradiation for 2 min. Cells were gently lysed in ice-cold IP lysis/Wash Buffer (Thermo Fisher Scientific) supplemented with Halt Protease and Phosphatase Inhibitor Cocktails (Roche). Cell lysate (50 μg proteins) was incubated overnight with anti-GSDMD antibody (10 μg/ml). 20 μl of protein A/G agarose (Santa Cruz Biotechnology, Inc) were added for additional 6 h. The beads were then washed five times with 10 mM Tris–HCl/20 mM NaCl. The associated complexes/proteins were released from the immunocomplexes by incubation for 5 min at 100 °C with 2% SDS. The dissociated complexes were collected, and 5 ng of DNA was blotted on a positively charged nylon membrane using the Bio-Dot Microfiltration System, and then cross-linked by UV irradiation for 3 min. The membranes were blocked with 5% BSA in TBST for 1 h at room temperature. Primary antibody 8OHdG was diluted 1:200 in 5% BSA and incubated overnight. After washing in TBST for 3 times, the membranes were incubated for 2 h at room temperature with Poly HRP-conjugated anti-mouse IgG.

## Flow cytometry analysis

The cells were stained with Zombie (BioLegend) in PBS for 30 min at 4 °C. Then the cells were stained with Fc blocker in FACS buffer (1× PBS with 0.5% BSA, 2 mM EDTA). Then the cells were resuspended in 50 μl FACS buffer with diluted indicated flow cytometry antibodies and incubated for 30 min at 4 °C. For ROS detection, bone marrow neutrophils were seeded in 6-well plates. Cells were pretreated with G-CSF (40 ng/ml) and IMT1 (10 μM) for 24 h, or pretreated with CsA (1 μM) plus VBIT-4 (10 μM) for 2 h. Then the cells were treated with IFN-γ (100 ng/ml) and LS (10%) for 12 h. Cells were then stained with Dihydrorhodamine 123 (ThermoFisher, D23806) with 1640 medium for 30 mins at 37 °C. The cells were washed twice by FACS buffer and analyzed by flow cytometry. Data were acquired with BD FACSDiva (v8.0.2) on a BD Fortessa X20 (BD Biosciences) and analyzed using FlowJo software (Tree Star, Inc.).

## ELISAs

Serum were isolated from whole blood using serum collection tubes and centrifuge at 3000 r.p.m for 10 min. Serum were collected and initially diluted to 1:10. Commercially available ELISAs kits were used for detection of anti-ANA (Alpha Diagnostic, San Antonio, TX), anti-ssDNA (Alpha Diagnostic, San Antonio, TX), anti-RNPs (Arotec, Wellington, New Zealand), anti-Sm (Arotec, Wellington, New Zealand), IL-1β (RayBiotech, GA, USA), 8OHdG (Cell Biolabs, CA, USA), IFN-α (Sangon Biotech, Shanghai, China) and IFN-β (Sangon Biotech, Shanghai, China). For detection of MPO-DNA and NE-DNA complexes[71], anti-human MPO antibodies (abcam ab25989, 1:1000) and anti-human NE antibodies (ab21595, 1:1000) were diluted with PBS and coated onto 96-well microtiter plates (75 μl per well) overnight at 4 °C. After blocking in 1% BSA (125 μl per well) for 90 min at room temperature, serum was added per well, followed by incubation overnight at 4 °C. The plate was washed 5 times, followed by the addition of Pico Green from Quant-iT Pico Green dsDNA assay kit and detected following the instructions (Invitrogen).

## Cytotoxicity assay

After stimulation as indicated, supernatant was collected and LDH release was quantified using CytoTox 96 Non-Radioactive Cytotoxicity Assay (Promega) according to the manufacturer's instructions. The calculation of percentages of cytotoxicity was analyzed on maximum LDH release from cells treated with 1% Triton X-100.

## Immunofluorescence

Murine or human neutrophils were seeded at $5 \times 10^5$ cells/ml on confocal dishes pre-coated with 0.001% poly-L-Lysine overnight. Cells were stimulated and treated as indicated. Neutrophils were fixed with 2% paraformaldehyde (PFA) at 37 °C for 15 min. After washing with PBS, cells were permeabilized by 0.1% Triton X-100 at 4 °C for 10 min. After being washed again, cells were blocked by 3% goat serum and 3% FBS at 4 °C for 1 h. The cells were incubated with indicated primary antibodies at 4 °C overnight. Then the cells were gently washed with PBS twice. Cells were incubated with secondary antibodies at room temperature for 2 h and were incubated with Hoechst or Sytox Green at room temperature for 10 min. For mtDNA detection, $2 \times 10^6$ cells/ml cells were seeded same as above. Cells were added with MitoSox at 100 nM (Invitrogen) for 15 min at 37 °C and then fixed in 2% PFA for 15 min at 37 °C for further staining. After washing with PBS, cells were permeabilized by 0.1% Triton X-100 at 4 °C for 10 min. After being washed again, cells were blocked by 3% goat serum and 3% FBS at 4 °C for 1 h. Anti-8-OHDG and anti-TOMM20 was added overnight at 4 °C, followed by incubation with secondary antibodies for 3 h at 4 °C. Cells were stained with Hoechst, DAPI or Sytox Green for 15 min. Kidneys sections from frozen OCT embedded tissues were fixed with 1:1 acetone/methanol and stained with IgG (FITC, Jackson ImmunoResearch) and anti-complement C3 (abcam) antibodies. Images were captured with FV3000 confocal system (Olympus) and data were analyzed with Imaris (Bitplane, V9.5) and ImageJ (NIH, V2.0.0).

## Mitochondrial membrane potential detection

Neutrophils were isolated from WT, pristane-induced WT and pristane-induced $Gsdmd^{-/-}$ mice. Cells were seeded at $2 \times 10^6$ cells/ml on the glass-bottom dishes pre-coated with 0.001% poly-L-Lysine overnight, and were cultured in RPMI supplemented with 10% FBS for 1 h at 37 °C. After supernatant removal, cells were incubated with JC-1 (Beyotime, Jiangsu, China) for 20 min at 37 °C following manufacturer's instruction. After washed by PBS twice, cells were incubated with Hoechst for 5 min at 37 °C. Images were performed on Olympus FV3000 Laser Scanning Microscope. Cells with low mitochondrial membrane potential showed green fluorescence and red fluorescence with high membrane potential.

## Assessment of renal disease

Mice were caged in metabolic cages and urine was collected over a 24 h period. Enzyme-linked immunosorbent assay kits were used to measure urine albumin and creatinine levels (albumin, Bethyl Laboratories, Houston, TX; creatinine, Exocell, Philadelphia, PA). The UACR ratio is expressed as micrograms of albumin to milligrams of creatinine.

## PI staining

Bone marrow neutrophils were seeded at $2 \times 10^6$ cells/ml on confocal dishes pre-coated with 0.001% poly-L-Lysine overnight. Cells were stimulated and treated as indicated. Neutrophils were treated with 2 μl PI (40302-B, Yeasen, Shanghai, China) in 200 μl PBS, reaction at room temperature and away from light for 10-15 min. Then the cells were washed with PBS for twice. Images were performed on Olympus FV3000 Laser Scanning Microscope.

## mtDNA isolation

Mitochondria were isolated using Cell Mitochondria Isolation Kit (Beyotime Biotechnology, Shanghai, China) according to manufacturer's

instructions. Briefly, the cells were gently resuspended in ice-cold PBS and centrifuged at $600 \times g$ (4 °C) for 5 min. The mitochondrial separation reagent (2 ml) with PMSF (100 mM) was added at a cell density of 50 million cells. The cells were placed on ice for 15 min and then transferred to a Dounce homogenizer (Kimble/Kontes, Vineland, NJ) and disrupted 15 times on ice. The homogenate was centrifuged at $600 \times g$ (4 °C) for 10 min to remove unbroken cells and nuclei. The supernatant was carefully transferred to another centrifuge tube, and centrifuged at $11000 \times g$ (4 °C) for 10 min. The supernatant was carefully removed, and the separated cell mitochondria were collected. MtDNA was isolated using Mitochondrial DNA Extraction Kit (Phygene, Fujian, China) according to manufacturer's instructions. One hundred and ten microliters DNase I (2 ng/µl) was mixed with mitochondria at 37 °C for 10 min to remove nuclear DNA. After wash and centrifuge at $12,000 \times g$ (4 °C) for 5 min. The pellet was resuspended and lysed with mitochondrial lysis buffer to release mitochondrial DNA. After enzyme mix treatment and ethanol precipitation, mitochondrial DNA was pelleted and resuspended in TE buffer for further analysis.

### Quantification of extruded mtDNA

The release of extracellular mtDNA were detected as described previously[21]. Two nanograms of DNA was isolated from cell-free supernatants by using mammalian genomic DNA extraction kit (Beyotime). The DNA were subjected to Real-Time PCR with SYBR green PCR master Mix (Vanzyme) and 0.5 µM of mtDNA primers described above. The results were expressed as relative mtDNA copy number.

### GSDMD oligomerization assay

Native gel immunoblot was performed as previously described[10,15]. $4 \times 10^6$ bone marrow neutrophils were seeded in 6-well plates. IFN-γ (100 ng/ml) and lupus serum (10%) were added for 12 h. Cells were harvested by 250 µl of 5 × SDS loading buffer. Samples were split equally into two tubes. And 75 µl of dH₂O or TCEP was added to generate non-reducing and reducing Western blotting samples. After heating to 65 °C for 10 min. Samples were electrophoresed through a 4–16% NativePAGE Bis-Tris gel (Invitrogen) in NativePAGE running buffer (Invitrogen) at 4 °C and 150 V. Proteins were then transferred to a PDVF membrane at 0.2 A for 2 h in NativePAGE transfer buffer (Invitrogen) for immunoblotting.

### Cell-free GSDMD cleavage by recombinant caspases

GSDMD cleavage was performed according to a previous report[8]. In short, 5 mg of purified recombinant human GSDMD was incubated with 0.1 mg of caspase-4 and LPS or 0.1 mg of active caspase-1 in a 25 ml reaction buffer (50 mM HEPES (pH 7.5), 3 mM EDTA, 150 mM NaCl, 0.005% Tween-20). Human mtDNA was obtained from mitochondria isolated from HEK293T cells using the protocol as previously described[72]. The mtDNA was oxidized by UV irradiation (250 mJ/cm²) according to previously published methods[73]. Human mtDNA or Ox-mtDNA was added at 20 nM. The reaction was incubated at 37 °C for 1 h. The GSDMD was separated through non-reducing SDS-PAGE gel.

### Protein labeling and microscale thermophoresis (MST) analysis

The binding affinity of GSDMD and mtDNA, GSDMD and Ox-mtDNA, GSDMD-N and mtDNA and Ox-mtDNA were measured at 25 °C in a binding buffer (PBS containing 0.05% Tween-20) by Microscale Thermophoresis using MONOLITH NT.115 as previously described[74]. One hundred microliters His-GSDMD, His-GSDMD-N peptide and His-GSDMD-N mutant peptide were added at a concentration of 160 nM and incubated with 100 µl 100 nM RED-tris-NTA 2nd Generation dye (NanoTemper) in the dark for 30 min. After incubation, the labeled GSDMD was mixed 1:1 with mtDNA or Ox-mtDNA in a twofold dilution series from 30 nM to 100 µM for the measurement. The samples were loaded into NanoTemper Monolith NT.115. Kd values were calculated

using the mass action equation and NanoTemper software, the data were plotted by the GraphPad Software.

### Statistics and reproducibility

All experiments were performed at least three times unless specifically stated. Information about statistical details and methods is indicated in the figure legend. Statistical analysis was performed using GraphPad Prism 8.0. Data are expressed as mean ± SD (standard deviation). Unpaired two-tailed Student's $t$ test was used to analyze the differences between two groups. Comparisons among multiple groups were analyzed with one-way analysis of variance. Survival curves were estimated for each group, considered separately, using the Kaplan–Meier method and compared statistically using the log rank test. $P < 0.05$ was considered statistically significant.

### Reporting summary

Further information on research design is available in the Nature Portfolio Reporting Summary linked to this article.

## Data availability

All data needed to evaluate the conclusions in the paper are present in the paper and/or the Supplementary Materials. The RNA-sequencing data have been deposited in Gene Expression Omnibus (GEO) of NCBI under the Accession number GSE173767 (https://www.ncbi.nlm.nih.gov/geo/query/acc.cgi?acc=GSE173767). Source data are provided with this paper.

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

## Acknowledgements

We thank Core Facility of Basic Medical Sciences, Shanghai Jiao Tong University School of Medicine. This work was supported by grants from The Ministry of Science and Technology of China (2020YFC2002800 to Jing Wang and X.Z.), the National Natural Science Foundation of China (81822020, 92042304, 31872737 to Jing Wang, 81700601 to J.Y., 81901636 to N.M.), the Natural Science Foundation of Shanghai (21ZR1456100 to N.M.).

## Author contributions

N.-J.M. and Z.-N.W. designed and performed experiments, analyzed data and interpreted results. Q.-L.W., H.-Y.X., N.-H.Y., Y.-Z.W., R.H., Y.-Z.W., Jin W., K.-P.Y., H.-X.K., Y.W., Q.-Y.H., W.-J.B., and Y.-Y.W. helped with the experiments. F.W., F.-B.L., F.G., Z.-Y.L., X.-L.Z., L.-M.L. and J.-Y.Y. provided study materials. J.-Y.Y. provided clinical samples. L.-M.L. and Jing W. supervised the study and wrote the manuscript.

## Competing interests

The authors declare no competing interests.
