## [Peer Review File · Nature Communications]

Control of gasdermin D oligomerization by oxidized mitochondrial DNA in systemic lupus erythematosusREVIEWER COMMENTS

Reviewer #1 (expertise in SLE, innate immunity in renal disease):

Xu et al. describe a novel role of mtDNA, and its oxidized form (oxmtDNA), in inducing GSDMD oligomerization and promoting pore formation in neutrophils upon treatment with IFN γ and serum from patients with SLE (LS). Moreover, they correlate SLE severity with cleaved GSDMD and levels of mtDNA from human PBMC-derived neutrophils and serum, respectively. They further confirm the role of GSDMD in SLE progression by genetic ablation or pharmacologically inhibiting GSDMD in a mouse model. The experimental design is accurate, and the findings are relevant including the translational data. The rationale and text are coherent and well-structured. However, there are some points that require causal evidence.

Major:

1) GSDMD pore formation is regulated downstream the inflammasome, which activate caspase proteins and subsequent GSDMD cleavage. The authors showed this in Figure 4, where GSDMD is activated through FC/Serpinb1/caspase-1 and 11; and Figure 5E where caspase-1^{-/-} and caspase-11^{-/-} neutrophils showed reduced GSDMD cleavage upon IFN γ and LS treatment. The authors performed a cell-free assay, which ultimately proves the role of oxmt-DNA in GSDMD oligomerization, with LPS + Caspase-4. Does the same occur with Caspase-1? The LS media used in vitro could contain GSDMD cleavage and oligomerization inducers other than mtDNA, such as active caspases. Could the authors prove that the effect is mtDNA specific? For instance, is the effect of LS abrogated when treated with DNases?

2) The human data is correlative while the animal data is based on total GSDMD inhibition. In both cases, there is not conclusive and causal evidence showing that oxmt-DNA exacerbates disease progression. Does mtDNA immunoprecipitate with GSDMD in the mouse kidneys or in mouse/human neutrophils? Could the authors measure the levels of mtDNA in the animal models as they do for human SLE? Could the authors inject mtDNA or DNases to accelerate/improve disease progression in mice?

3) The authors show that ox-mtDNA and, to a lesser extent, mtDNA interact with GSDMD (figure 7F). Moreover, scavenging ROS decreases GSDMD oligomerization (Figure 6D). Co-immunoprecipitation experiments with GSDMD-N pull down mtDNA in neutrophils treated with LPS+Nigericin+H₂O₂ but not in neutrophils treated with LPS+Nigericin (Figure 7A). It is therefore not clear whether non-oxidized mtDNA interacts with GSDMD, and whether this is relevant for GSDMD oligomerization. Are ROS essential in GSDMD oligomerization because they oxidize mtDNA? Experimental evidence is needed to address these points.

Minor:

- 1) Caspase-1 and -11 Western Blots have many bands: it is difficult to distinguish which band is the active one. Could you please mark it?
- 2) Gasdermin-N WBs appear duplicated in two different exposures. Are these the same samples with different exposures? Could you please show just the clearest exposure?
- 3) Figure 7 C and D. Is the first column (control) treated with H₂O₂? (Exactly as the second one?) or the control is not treated with H₂O₂?
- 4) Line 81. Spell out DAMP
- 5) Line 349. Typo in "Subsequently there are studies showed..."Line 348.

Reviewer #2 (expertise in gasdermins, innate immunity):

- What are the noteworthy results?

In this paper the authors propose that in SLE mitochondrial DNA directly interacts with cleaved GSDMD in neutrophils to help this protein oligomerize into pores to drive inflammation. GSDMD could be a novel therapeutic target in this disease if these data in this paper are confirmed. Key data supporting this hypothesis include functional analysis, structural analysis and work in both

mouse and human cells including clinical samples.

- Will the work be of significance to the field and related fields? How does it compare to the established literature? If the work is not original, please provide relevant references.

Potentially this work is interesting and original. If GSDMD is important in SLE this offers potential for the development of new therapies. The suggestion that mitochondrial DNA directly targets GSDMD to facilitate its oligomerization is also novel. Mitochondrial DNA has been suggested to activate NLRP3 in the past (as acknowledged by the authors in reference 35) and this potential mechanism to drive GSDMD oligomerization needs to be carefully distinguished from the results presented here. Neutrophils, unlike macrophages, have a controversial relationship with GSDMD in the context of cell death which needs to be acknowledged, explained and compared to the results seen here. The relationship of the data in this MS to doi: 10.1096/fj.202100085R should be discussed.

- Does the work support the conclusions and claims, or is additional evidence needed?

1. What drives GSDMD, caspase 1 and caspase 11 upregulation of expression?

2. All imaging needs proper quantification. This includes the number of cells/tissues analysed, the number of fields of view assessed per experiment, how many independent experiments were done so when the authors say an n of 3 what does this mean? 3 cells from 3 people, 3 fields of view from 3 cells etc? How common are these events: "indicating GSDMD-dependent cell death in lupus mice. These results indicated GSDMD-dependent neutrophil extracellular DNA release in lupus mice"? Again quantification of the imaging data will answer this question.

3. Fig 1. The authors need to explain why they focused on GSDMD? This protein together with caspase-1 and caspase-11 (labelled 4 here) are upregulated in a mouse model of lupus, but the TLR cassette is even more upregulated than GSDMD, caspase 1 or caspase 4 (should be 11)? What is most important GSDMD or TLRs in driving pathogenesis in the model?

4. Does propidium iodide get taken up into the cells showing pores are forming? This could be compared between SLE, HV and used in mouse cells to show functional pore formation and confirm pyroptotic cell death

5. Fig 3. This referee may have missed this, but were the reverse controls for the "we isolated neutrophils from HVs and cultured these cells with the serum of SLE patients" done so neutrophils from SLE cultured with serum from HV? If the pathology relies upon upregulation of GSDMD, caspase 1 and caspase 4 its not completely clear why the HV neutrophil plus SLE serum experiment works?

6. The statement "Gasdermin pores have lytic and non-lytic functions that control the release of intracellular contents" is not really accurate as the cell lysis is driven by NINJ1 so this phrase needs to be rephrased

7. Fig 5 why has IFN now been introduced into the experimental protocol along with serum from lupus to cleave GSDMD? What is the IFN doing? Do the authors think IFN is altering GSDMD expression? The KO mice data in this experiment are not very convincing

8. Whats cleaving GSDMD upstream of caspase 1 or 11? What is the mechanism here by which Serpinb1 stops caspase 1 or 4 activation and GSDMD cleavage? Can the SLE pathology be ameliorated by an NLRP3 inhibitor for example if this is the protein responsible? The authors have quoted that ox-mitochondrial DNA has been suggested to be an activator of NLRP3 so it is unclear why this has not been investigated?

9. Is the OxMtdNA controlled for LPS contamination? How did the authors ensure there was no LPS contamination from E coli?

- Are there any flaws in the data analysis, interpretation and conclusions? - Do these prohibit publication or require revision?

See comments above

- Is the methodology sound? Does the work meet the expected standards in your field?

The methodology seems solid.

- Is there enough detail provided in the methods for the work to be reproduced?

The methodology description is reasonable

Reviewer #3 (expertise in type-I interferons in autoimmunity, inflammation):

In experiments utilizing MRL.lpr mice, pristane-induced lupus (PIL) and peripheral blood cells obtained from human SLE patients, the authors report that gasdermin D (GSDMD) becomes oligomerized and activated resulting in inflammation and cell death. They report the effects of

Gsdmd deficiency as well as after treatment of lupus mice with a non-selective small molecule inhibitor of GSDMD, disulfiram (DSF), in vitro and in vivo to show relevance. Observing beneficial effects, they propose that these inhibitors be considered for lupus therapy. Inflammasome activation has been previously reported in murine models of SLE (reviewed in Kahlenberg & Kaplan, *Curr Opin Rheumatol*, 2014) and links between inflammasome activation, mitochondria and cGAS-STING activation have been previously published (e.g. Aareberg et al, *Mol Cell*, 2019). The most original aspects of the study are i) the experiments showing GSDMD oligomerization as a consequence of released oxidized mitochondrial DNA (mt DNA) which the authors show directly binds to GSDMD and ii) the beneficial effects of Gsdmd deficiency on murine lupus as well as the therapeutic effect of a non selective inhibitor, disulfiram (DSF) in the two mouse models of lupus.

1. This is a very ambitious study resulting in much breadth but insufficient depth. Another general problem is that in many experiments only 3 samples are tested. The numbers should be increased to 5-6 in each group for most experiments.

2. Much of the work focuses on the effects of lupus serum on mouse or human neutrophils in the generation of GSDMD-N. The manuscript could be strengthened by a more intense and thorough study of the lupus mice and spontaneous abnormalities in human SLE cells.

3. Significant limitations of this study include a lack of rigor considering that there are too many experiments shown with single fluorescence or western blot without summary data to convince reviewers that 'representative examples' are reproducible findings. This is compounded by the fact that results in SLE patients are interwoven with the mouse data (often two different models) such that it is difficult to ascertain the extent to which the paradigm applies to human SLE.

There are a number of additional issues in the individual experiments.

Fig. 1. The rationale for investigating the inflammasome pathway is not so clear since Fig. 1A shows very high upregulation of IRF3, IRF7 and Rel A with little or modest upregulation of Gsdmd and Casp4. Please explain the relevance of IRF3, IRF7 and Rel A in these experiments.

Fig. 1B. How do the authors exclude that GSDMD is more activated in macrophages than neutrophils? Figs. 1B-D and S1 B-D need rigorous quantification – interpretation of "representative figs" is not adequate.

What are the "Controls" in Fig. 1B-D?

Fig. 1J. Why are two bands labelled GSDMD-N? Also seen in Fig. 5C & E.

key Figure 2 G missing

Fig 3 & 4. These experiments essentially confirm other studies (see Chen, Demarco, Broz *EMBOJ*, 2019) indicating GSDMD can be responsible for NET formation, although in a lupus context.

In Fig. 3B, D, & L show differences in NET formation when GSDMD is lacking or inhibited. However, in most of these studies, there is not complete inhibition. The authors need to address this point - does this mean GSDMD contributes but does not fully explain NET formation? Fig. 3O, why was IFN-g added to the experiment? The red and purple are near impossible to see the overlap – which needs proper quantitation as mentioned for Fig. 1. The bottom panel needs to be labeled.

Fig. 4 E-G microscopy pictures don't always correspond to quantitation - DSF looks much less efficient at blocking 8OHdG in Fig 4E whereas DSF and MT look the same in Fig. 4G.

Fig. 5G, 4th lane looks more loaded than lane 5 (IgG depleted). This experiment needs proper quantitation (protein of interest / GAPDH ratio) over multiple experiments. Same with Fig. 4I – whereas in Fig. 5I the differences in serpin is clear (at least in this one experiment).

Fig. 6 contains necessary results for the conclusions yet only 3 data points are shown for key experiments – especially Fig. 6 G-L. The conclusions should be based on at least 5-6 mice per group. Fig. 6A, why is there no background in the Gsdmd KO lanes? What is the explanation for the discrepancy in the short term vs long term treatment with mitotracker? Fig. 6K is not adequately controlled. What about the addition of non-oxidized DNA. A dose titration effect would be more convincing.

Fig. 7C-D – transfection efficiencies of the constructs need to be shown by WB. Fig. 8 H-I need proper quantitation in at least 5 mice per group. Fig. 7, What does the Y axis "relative expression of IgG" mean? Fig. 7 - what was the result with H₂O₂ alone? Can statistical analysis be applied to a comparison between Figs E and F?

Fig. 8 The results from floxed GSDMD in neutrophils are an important addition to the experiments and help support the conclusions. The red and blue labeling in Figs 8C D-G seem opposite. The results in Fig. 8 C need better explanation in the figure legend –the green line above Gsdmd is higher in the knockout suggesting higher expression?

Fig. 9 – Since MRL.lpr were treated at 6 weeks, this is more of a prevention than a treatment study. The results indicate partial improvement. IL-18 has been implicated in lupus pathogenesis in this strain and is also regulated by the inflammasome. What was the effect of DSF on IL-18? Fig. 9K needs proper quantitation in at least 5 mice per group.

Other points

Confusing nomenclature p.3 e.g. (Fig. S1, E and F), should read Fig. 1 E, F and Fig. S1)

I. 119 pyroptosis is not "new form of cell death" having been described more than 20 years ago

I. 151 the authors need to state when using mouse vs human

I. 177 DSF application in vitro needs viability controls

I.233 too confusing to mix mouse and human especially when using human reagents on mice

I.284 - these are not typical ISGs

I.301 looks like myeloid reduced

Fig. S4 should have specific nephritis score also quantitation of IgG in the kidney. Sometimes the authors report proteinuria and sometimes UCAR. Kidney descriptions should be consistent

We are grateful for the valuable comments and suggestions from the reviewers of this manuscript. The rigorous review has helped us immensely improve the manuscript. The suggestions and comments have been closely followed and revisions have been made accordingly. We sincerely appreciate the valuable comments and suggestions from the reviewers. Please see below, our point-by-point response to comments.

RESPONSE TO REVIEWER COMMENTS

Reviewer #1 (expertise in SLE, innate immunity in renal disease):

Xu et al. describe a novel role of mtDNA, and its oxidized form (ox mtDNA), in inducing GSDMD oligomerization and promoting pore formation in neutrophils upon treatment with IFN γ and serum from patients with SLE (LS). Moreover, they correlate SLE severity with cleaved GSDMD and levels of mtDNA from human PBMC-derived neutrophils and serum, respectively. They further confirm the role of GSDMD in SLE progression by genetic ablation or pharmacologically inhibiting GSDMD in a mouse model. The experimental design is accurate, and the findings are relevant including the translational data. The rationale and text are coherent and well-structured. However, there are some points that require causal evidence.

Major:

1) GSDMD pore formation is regulated downstream the inflammasome, which activate caspase proteins and subsequent GSDMD cleavage. The authors showed this in Figure 4, where GSDMD is activated through Fc/Serpinb1/caspase-1 and 11; and Figure 5E where caspase-1^{-/-} and caspase-11^{-/-} neutrophils showed reduced GSDMD cleavage upon IFN γ and LS treatment. The authors performed a cell-free assay, which ultimately proves the role of oxmt-DNA in GSDMD oligomerization, with LPS + Caspase-4. Does the same occur with Caspase-1? The LS media used in vitro could contain GSDMD cleavage and oligomerization inducers other than mtDNA, such as active caspases. Could the authors prove that the effect is mtDNA specific? For instance, is the effect of LS abrogated when treated with DNases?

Response: We thank the reviewer for the valuable suggestions. Based on your concern regarding caspase-1 in GSDMD cleavage and oligomerization with mtDNA, we

performed an experiment in which recombinant active caspase-1 was incubated with purified human GSDMD protein, and then human mtDNA was isolated from HEK293T cells. MtDNA or Ox-mtDNA was added separately. The data showed that both mtDNA and Ox-mtDNA promote GSDMD oligomerization in the caspase-1/GSDMD cell-free system (Fig R1). These results indicated that the system of caspase-1/GSDMD just as effective of LPS/Caspase-4/GSDMD system in exploring mtDNA-dependent GSDMD oligomerization.

Fig R1 (A) Western blot analysis of GSDMD oligomer under non-reducing conditions. Purified GSDMD was incubated with active caspase-1 *in vitro*, and then mtDNA and Ox-mtDNA were separately added to the system. (B) Quantitative analysis of GSDMD oligomer/GSDMD monomer. n = 5 pooled from 5 independent experiments. (Added in Fig. S3, A and B).

In response to your comments on how to prove the specific role of mtDNA, which is the most important question in our manuscript. Firstly, murine neutrophils were pretreated with inhibitor of mitochondrial transcription (IMTs) that cause a dose-dependent decrease in the levels of mitochondrial transcripts and gradual depletion of mtDNA (Nature, Dec; 588(7839): 712-716). Secondly, cells were treated with a combination of the inhibitor of VBIT-4 (the outer mitochondrial membrane pore VDAC1 oligomerization inhibitor) and CsA (binds cyclophilin D and inhibits Ca²⁺ regulated mPTP opening) to fully suppress the release of mtDNA from the mitochondria to the cytoplasm (Immunity, Aug 9; 55(8): 1370-1385.e8.). The results showed that both IMT1 and CsA with VBIT-4 inhibit GSDMD oligomerization (Fig R2). The results indicated that mtDNA did participate a role in GSDMD

oligomerization.

Fig R2 (A) Western blot of GSDMD from the neutrophils under non-reducing conditions. Murine bone marrow neutrophils were isolated. Cells were pretreated with IMT1 (5 μ M) or CsA (1 μ M) with VBIT4 (10 μ M). Then the cells were treated with IFN- γ plus LS. (B) Quantitative analysis of GSDMD oligomer/GAPDH. n = 5 biological replicates pooled from 3 independent experiments. (Added in Fig 6, Q and R).

Finally, we added DNase I into the serum from lupus mice to digest DNA in the serum. The results showed that GSDMD oligomer was reduced after DNase I treatment, suggesting that nuclear DNA or mtDNA in serum can also promote GSDMD oligomerization (Fig R3). Since neutrophil phagocytosis can be activated by FcR (mBio, Oct 4;7(5):e01624-16) and Complement C3 (Adv Mater, Aug;34(34):e2203477), we hypothesis that mtDNA from serum may be phagocytosed by neutrophils to further promote GSDMD oligomerization.

Fig R3 (A) Western blot of GSDMD from the neutrophils under non-reducing conditions. Murine

bone marrow neutrophils were isolated. Lupus serum was pretreated with DNase (0.1 mg/ml) and then the cells were treated with IFN- γ plus LS. (B) Quantitative analysis of GSDMD oligomer/GAPDH. n = 5 biological replicates pooled from 3 independent experiments. (Added in Fig S3, E and F).

2) The human data is correlative while the animal data is based on total GSDMD inhibition. In both cases, there is no conclusive and causal evidence showing that oxmtDNA exacerbates disease progression. Does mtDNA immunoprecipitate with GSDMD in the mouse kidneys or in mouse/human neutrophils? Could the authors measure the levels of mtDNA in the animal models as they do for human SLE? Could the authors inject mtDNA or DNases to accelerate/improve disease progression in mice?

Response: We appreciate the questions that the role of *in vivo* mtDNA in lupus disease. As you suggested, we performed *in vivo* experiments in MRL/lpr mice with mtDNA and DNase treatment, and we further compared these effects with DSF and NLRP3 inhibitor MCC950 in MRL/lpr mice.

Immunoprecipitation of GSDMD and mtDNA were carried out in peripheral blood neutrophils from healthy volunteers and SLE patients. Western blot and dot blotting analysis showed that both TOMM20 and 8-OHdG were immunoprecipitated by GSDMD antibody in neutrophils from SLE patients (Fig R4), indicating a direct interaction between GSDMD and mtDNA in neutrophils from SLE patients.

Fig R4 (A) Cell lysates were immunoprecipitated with anti-GSDMD, and immunoprecipitates were

separated on SDS-PAGE and probed with antibodies specific for GSDMD and TOMM20, or spotted on a nitrocellulose membrane, UV crosslinked and probed with antibodies specific for 8OHdG. (B) Quantitative analysis of 8-OHdG/GSDMD and TOMM20/GSDMD. n = 6 HV or SLE patients pooled from 3 independent experiments. (Added in Fig 7, D and E).

In response to your question regarding the levels of mtDNA in the animal models, we measured the levels of mtDNA in PIL and MRL/lpr mice. The results showed that the level of mtDNA was significantly reduced in *Gsdmd*^{-/-} mice after pristine treatment or DSF treatment in MRL/lpr mice.

Fig R5 (A) qPCR analysis of D-loop in serum from WT and *Gsdmd*^{-/-} mice after pristine treatment.

(B) qPCR analysis of D-loop in serum from *Gsdmd*^{fl/fl} and *Gsdmd*^{fl/fl}S100A8-Cre mice after pristine treatment. (C) qPCR analysis of D-loop in serum from MRL/lpr and MRL/lpr with DSF-treated mice. n = 6 biological replicates pooled from 2 independent experiments (Added in Figs S5I, S7E, and 9J).

Finally, we injected mtDNA (0.1 µg/mice) through caudal vein or DNase I (100 U/mice) through intraperitoneal to accelerate/inhibit disease progression in MRL/lpr mice. We also orally administered NLRP3 inhibitor MCC950 and DSF to MRL/lpr mice at 10 weeks for 10 weeks. DSF and DNase I treatments suppressed disease-related increases in splenic and lymph node weight in MRL/lpr mice (Fig. R6A-C). mtDNA increased the weight of spleen and lymph node while MCC950 showed no effect (Fig. R6, A-C). DSF and DNase I treatments reduced serum levels of anti-dsDNA (Fig. R6D), anti-RNPs (Fig. R6E), and anti-Sm (Fig. R6F) autoantibodies in MRL/lpr mice. mtDNA increased anti-dsDNA, anti-RNPs, and anti-Sm levels while MCC950 had no

effect (Fig. R6D-F). DSF and Dnase I treatments also reduced serum levels of IL-1 β and IL-18 in MRL/lpr mice (Fig. R6G and H). mtDNA increased serum IL-1 β and IL-18 levels while MCC950 had no effect (Fig. R6G and H). DSF, Dnase I, and MCC950 treatments all decreased renal deposition of immune complexes (IgG and complement C3) (Fig. R6I and J), while injection of mtDNA promoted IgG and C3 deposition. Kidney histopathological analyses revealed severe glomerular, crescent formation, increased mesangial matrix, tubular atrophy, and diffuse perivascular and interstitial mononuclear cell infiltration in MRL/lpr mice, all of which were alleviated after DSF, Dnase I, or MCC950 treatments. DSF, Dnase I or MCC950 treatments reduced kidney function dysregulation in MRL/lpr mice, which was measured using the urine albumin-to-creatinine ratio (UACR) (Fig. R6L).

Fig R6 (A) Image of spleen and lymph nodes. (B, C) Quantitative analysis of spleen and lymph node weight from indicated groups. (D-H) ELISA analysis of serum anti-dsDNA, anti-RNPs, anti-Sm, IL-1 β , and IL-18 levels from indicated groups. (I) Immunofluorescence analysis of IgG and C3 of the kidney from indicated groups. H&E staining of kidney sections from indicated groups. (J-L) Quantitative analysis of the intensity of IgG and C3, the number of crescents, and UACR from indicated groups. Each field of view represents one mouse. n = 6 biological replicates pooled from 2 independent experiments. * $P < 0.05$, ** $P < 0.01$, *** $P < 0.001$, **** $P < 0.0001$. (**Only shown in the rebuttal**).

3) The authors show that ox-mtDNA and, to a lesser extent, mtDNA interact with GSDMD (figure 7F). Moreover, scavenging ROS decreases GSDMD oligomerization (Figure 6D). Co-immunoprecipitation experiments with GSDMD-N pull down mtDNA in neutrophils treated with LPS+Nigericin+H₂O₂ but not in neutrophils treated with LPS+Nigericin (Figure 7A). It is therefore not clear whether non-oxidized mtDNA interacts with GSDMD, and whether this is relevant for GSDMD oligomerization. Are ROS essential in GSDMD oligomerization because they oxidize mtDNA? Experimental evidence is needed to address these points.

Response: Thank you for your insightful comments. In response to your question regarding whether non-oxidized mtDNA interacts with GSDMD, we formed a cell-free

system of LPS+Caspase-4+GSDMD with mtDNA and Ox-mtDNA. The results showed that both mtDNA and Ox-mtDNA promote GSDMD oligomerization (Fig R7).

Fig R7 (A) Western blot analysis of GSDMD oligomer under non-reducing conditions. Purified GSDMD was incubated with LPS and caspase-4 *in vitro* and then mtDNA (20 nM) and Ox-mtDNA (20 nM) were separately added to the system. (B) Quantitative analysis of GSDMD oligomer/GSDMD monomer. n = 5 pooled from 5 independent experiments. (Added in Fig 6, M and N).

mROS has been shown to promote GSDMD oligomerization (10.1016/j.cell.2021.06.028.). In this manuscript, we added mitoTEMPO to suppress mROS, and the results showed reduced, but still residue GSDMD oligomer after IFN- γ plus LS treatment (Figure 6, E and F), indicating mROS-independent pathway in GSDMD oligomerization.

To further illustrate this problem, we added CsA and VBIT4 to inhibit mtDNA release from the mitochondria. Meanwhile, we measured ROS levels in all groups. It was found that although CsA plus VBIT4 treatment reduced GSDMD oligomerization (Fig R2), it did not affect the level of ROS (Fig R8). These results indicated that ROS was not required for GSDMD oligomerization when the mtDNA was restricted in the mitochondria. Thus, except for ROS, mtDNA also plays an important role in GSDMD oligomerization. In addition, mitochondrial ROS (mROS) participates in the loss of mitochondrial membrane potential of neutrophils, which promotes mtDNA release and mtDNA oxidization (Nat Med, Feb;22(2):146-53). Therefore, in our system, we think mROS is required to promote mtDNA release and oxidization, but ROS might also

promote GSDMD oligomerization in other mtDNA independent manner.

Fig R8 (A) Flow analysis of ROS from indicated groups. (B) Quantitative analysis of MFI of ROS. n = 5 biological replicates and 3 independent experiments. **Added in Fig. S3, G and H.**

1) Caspase-1 and -11 Western Blots have many bands: it is difficult to distinguish which band is the active one. Could you please mark it?

Response: Thanks for your suggestion, we have marked the active substrate of caspase-1 and caspase-11.

2) Gasdermin-N WBs appear duplicated in two different exposures. Are these the same samples with different exposures? Could you please show just the clearest exposure?

Response: Thank you for your comments. One of the duplicated GSDMD-N blots is pro-GSDMD and GSDMD-N. The other is specific exposure of GSDMD-N where pro-GSDMD is covered by tinfoil. These are the same samples. The cleaved GSDMD-N is hard to see when exposed together with pro-GSDMD (e.g., Fig 5, A and E). As you suggested, we have deleted the label of GSDMD-N in pro-GSDMD blots and only cited GSDMD-N in the second clearest exposure.

3) Figure 7 C and D. Is the first column (control) treated with H₂O₂? (Exactly as the second one?) or the control in not treated with H₂O₂?

Response: We sincerely apologize for our carelessness, and the first column is not treated with H₂O₂. We have corrected our mistakes in Figs 7C and D.

4) Line 81. Spell out DAMP

Response: We sincerely apologize for our carelessness, and we have spelled out damage-associated molecular pattern (DAMP) in the manuscript.

5) Line 349. Typo in “Subsequently there are studies showed...”Line 348.

Response: We sincerely apologize for our carelessness, and we have corrected this mistake.

Reviewer #2 (expertise in gasdermins, innate immunity):

- What are the noteworthy results?

In this paper the authors propose that in SLE mitochondrial DNA directly interacts with cleaved GSDMD in neutrophils to help this protein oligomerize into pores to drive inflammation. GSDMD could be a novel therapeutic target in this disease if these data in this paper are confirmed. Key data supporting this hypothesis include functional analysis, structural analysis and work in both mouse and human cells including clinical samples.

- Will the work be of significance to the field and related fields? How does it compare to the established literature? If the work is not original, please provide relevant references.

Potentially this work is interesting and original. If GSDMD is important in SLE this offers potential for the development of new therapies. The suggestion that mitochondrial DNA directly targets GSDMD to facilitate its oligomerization is also novel. Mitochondrial DNA has been suggested to activate NLRP3 in the past (as acknowledged by the authors in reference 35) and this potential mechanism to drive GSDMD oligomerization needs to be carefully distinguished from the results presented here. Neutrophils, unlike macrophages, have a controversial relationship with GSDMD in the context of cell death which needs to be acknowledged, explained and compared to the results seen here. The relationship of the data in this MS to doi: 10.1096/fj.202100085R should be discussed.

Response: We are grateful for your comments. Since NLRP3 plays an essential role in GSDMD cleavage, our main focus is to explore the role of NLRP3 in GSDMD oligomerization. Previously, mtDNA has been proven to activate NLRP3, which may promote GSDMD oligomerization. Therefore, we isolated bone marrow neutrophils from wild-type (WT) mice, and cells were pre-treated with MCC950. The results showed that MCC950 did not affect IFN- γ plus LS-induced GSDMD oligomerization, indicating that NLRP3 is not required for GSDMD oligomerization in neutrophils. Furthermore, the expression of NLRP3 inflammasome-related constituents was elevated in bone marrow-derived mesenchymal stem cells and monocytes/macrophages

in patients with SLE, but can be hardly detected in neutrophils (Biomed Pharmacother, Oct; 118: 109313). Thus, the role of NLRP3 may be more important in monocytes/macrophages but not neutrophils. And the activation of caspase-1 may dependent on serpinb1 or other inflammasomes but not on the NLPR3 inflammasome.

Fig R9. (A) Western blot analysis of GSDMD from neutrophils under non-reducing conditions. Murine bone marrow neutrophils were isolated. Cells were pretreated with MCC950 (7.5 μ M), followed by treatment with IFN- γ plus LS. (B) Quantitative analysis of GSDMD oligomer/GAPDH n = 5 biological replicates pooled from 3 independent experiments. (Added in Fig S3, C and D).

Regarding your interesting point that neutrophils have a controversial relationship with GSDMD in the context of cell death. In 2018, Sollberger et al. (Sci Immunol, Aug; 3(26): eaar6689) and Chen et al. (Sci Immunol, Aug; 3(26): eaar6676) identified that GSDMD-dependent membrane rupture promotes NETs after PMA treatment. Later, Karmakar et al. proved that GSDMD was predominantly associated with azurophilic granules and LC3 autophagosomes after LPS plus ATP treatment. We believe that these contradicting results are not unusual because the function of GSDMD depends on the strength or nature of the stimulus. In LPS plus ATP treatment, neutrophils released IL-1 β with no pyroptotic morphology (e.g., bubble formation). However, in an alternative model of LPS or PMA-induced NETosis, activation of caspase-11 promoted rapid production of cleaved N-GSDMD in amounts sufficient to permeabilize plasma membrane compartments for efficient externalization of DNA (Sci Immunol, Aug; 3(26): eaar6689). In lupus disease, neutrophils were sufficient to undergo NETs after immune complex treatment, which as we verified, was dependent on GSDMD-

mediated membrane rupture. In addition, we proved that GSDMD was cleaved by both caspase-1 and caspase-11, which are sufficient to cleave GSDMD into the GSDMD-N terminal.

Lastly, we added in discussion comparing our study with the latest study that gasdermins mediate the cellular release of mitochondrial DNA during pyroptosis and apoptosis (FASEB J, Aug;35(8): e21757.). “In consistent with a previous study in macrophage, GSDMD was identified to promote a fast mitochondrial collapse, cytosolic mtDNA accumulation, and mtDNA release from cells during pyroptosis and apoptosis. They further revealed that GSDMD pores were not big enough to allow mtDNA to be released from the cell, but are big enough to induce its release from the mitochondrial matrix. Our results showed that GSDMD knockout significantly suppressed extracellular mtDNA release after LS plus IFN- γ treatment. The discrepancy might be due to the different cell types and stimulations. It was identified that cell-free supernatants from healthy neutrophil cultures contain mtDNA in the absence of activation, while monocytes extrude negligible amounts of mtDNA (J Exp Med, May 2;213(5):697-713.). Furthermore, a recent study proved cytosolic Ox-mtDNA are not intact circular DNA, but 500-650 bp fragments (Immunity, Aug 9;55(8):1370-1385.e8.). So GSDMD is sufficient to promote extracellular mtDNA release in neutrophils after LS plus IFN- γ treatment.”

- Does the work support the conclusions and claims, or is additional evidence needed?

1. What drives GSDMD, caspase 1 and caspase 11 upregulation of expression?

Response: It has been reported that interferon regulatory factor 2 (IRF2) is required for GSDMD (Sci Signal, May; 12(582): eaax4917) and caspase-4 (EMBO Rep, Sep; 20(9): e48235) upregulation. Caspase-1 and caspase-11 were upregulated by the stimulation of LPS and IFN- γ (Science, Sep; 341(6151): 1250-1253; Nature, Oct; 526(7575): 666-671), in which the precise mechanism was the upregulation of IFN-stimulated response elements (ISRE) and IFN-stimulated gene (ISG). Therefore, in our *in vitro* study, we used IFN- γ to prime neutrophils to increase the precursors of GSDMD, caspase-1, and caspase-11.

2. All imaging needs proper quantification. This includes the number of cells/tissues analysed, the number of fields of view assessed per experiment, how many independent experiments were done so when the authors say an n of 3 what does this mean? 3 cells from 3 people, 3 fields of view from 3 cells etc? How common are these events: “indicating GSDMD-dependent cell death in lupus mice. These results indicated GSDMD-dependent neutrophil extracellular DNA release in lupus mice”? Again quantification of the imaging data will answer this question.

Response: Thank you for your good suggestion. We have quantified all the immunofluorescent staining data in our manuscript. We have provided the number of cells/tissues in each experiment. We have also provided imaging data from three independent experiments. We have added to at least five samples per group as the reviewer suggested. As you can see in Fig 2G, each symbol represents one movie captured from one mouse. The results showed that tdTomato⁺SG⁺ cells were significantly suppressed in *Gsdmd*^{-/-} mice after pristine treatment compared with control. We have also provided representative source movies from three independent experiments.

3. Fig 1. The authors need to explain why they focused on GSDMD? This protein together with caspase-1 and caspase-11 (labelled 4 here) are upregulated in a mouse model of lupus, but the TLR cassette is even more upregulated than GSDMD, caspase 1 or caspase 4 (should be 11)? What is most important GSDMD or TLRs in driving pathogenesis in the model?

Response: Thank you for your comments. Firstly, the most important reason we focused on GSDMD is that as a recently identified driving factor in pyroptosis, the role of GSDMD in lupus disease is largely unknown. GSDMD promotes cell pyroptosis and the release of inflammatory cytokines (IL-1 β , IL-18) and DAMPs (IL-1 α , HMGB1, tissue factor, mtDNA). From this aspect, it may be crucial to understand the role of cell pyroptosis and DAMP signaling in shaping autoimmune diseases such as SLE.

To the best of our knowledge, there are two reports regarding the role of GSDMD in lupus disease. In one report, Sun et al. found that GSDMD inhibitor disulfiram

alleviates pristane-induced lupus (*Cell Death Discov*, Sep; 8(1): 379). Our study verified the role of GSDMD in pristane-induced lupus mice using both GSDMD-ko and neutrophil-specific GSDMD-ko mice. We further reveal a direct interaction between Ox-mtDNA and GSDMD-N, and ox-mtDNA promotes GSDMD oligomerization. In the other report, Kaplan et al. identified the opposite role of GSDMD in murine lupus. However, they used imiquimod-induced lupus mice, which is not a canonical lupus mouse model. In another model, pristane, which is only injected for one week, was proposed; however, the autoantibodies could not be detected. In pristane-induced lupus mice, the autoantibodies can be detected at least after 4 months. Our study further identified the role of GSDMD in pristane-induced lupus mice at 7 months, which showed a good protective role of GSDMD in this canonical lupus mice model. In a recent review, pyroptosis and its role in autoimmune disease are purported to be a potential therapeutic target (*Front Immunol*, Ma; 13:841732). Thus, it is extremely important to fully identify the role and mechanism of GSDMD activation in lupus disease.

The human homologue of mouse caspase-11 is caspase-4 (*J Cell Biol*, May 1;149(3):613-22), whereas both caspase 4 and caspase 11 are referred to the same caspase in mouse. In Figure 1A, RNAseq data shows the official symbol for mouse Caspase 11(Casp4). Therefore, we kept Casp4 only in this figure but used caspase 11 in the rest of the manuscript to avoid confusion. We have clarified this in the manuscript. In addition, the reason that we picked caspase-1 and caspase-11 is that they are the upstream enzymes that cleave GSDMD into GSDMD-N.

We agree with the viewpoint that the TLR family is very crucial in SLE. TLRs are more important than GSDMD in driving SLE pathogenesis in our two murine models. TLR stimulation is believed to be an initial signal contributing to the activation and modulation of aberrant adaptive immune response (*Nature*, Jun; 465(7300): 937-941). More importantly, TLR7/9 and TLR7/8/9 oligonucleotide inhibitors had good effects in a preclinical study of patients with SLE (*Acta Pharmacol Sin*, Dec;36(12):1395-407). TLR7 and TLR9 are the most abundantly expressed in pDC and B cells. However, our study focused on neutrophils. Murine neutrophils are activated by TLR7 but not TLR9

agonists to secrete histamine (Immunol Lett, Dec; 141(1): 102-108.), and immune complexes containing TLR7 ligands induce neutrophils to undergo NETosis (Sci Transl Med, Mar; 3(73): 73ra20). Although TLRs play a more important role than GSDMD in SLE, GSDMD may be more important than TLRs in neutrophils and NETs. Finally, in our RNA-seq data of the kidney, the fold change of caspase-1, caspase-11 and GSDMD in the total kidney tissue were more robust than TLR9, indicating a predominant role of Caspase-1, 11/GSDMD in lupus nephritis (Fig R10). Hence, further studies are required to investigate the role of GSDMD in lupus nephritis. In addition, the function of GSDMD is attributed to its cleavage and oligomerization but not to its upregulation. Therefore, identifying the mechanism of GSDMD cleavage and its membrane pore-forming role in lupus disease is very important and urgent.

Fig R10. (A) Quantitative analysis of RNAseq data of indicated gene expression in the kidney from WT and pristane-induced lupus mice. n = 3 mice per group and two independent experiments. (**Only shown in the rebuttal**).

4. Does propidium iodide get taken up into the cells showing pores are forming? This could be compared between SLE, HV and used in mouse cells to show functional pore formation and confirm pyroptotic cell death.

Response: Thank you for your suggestion. According to previous studies, propidium iodide (PI) uptake reflects pore formation in pyroptosis (Cell Res, Sep; 26(9): 1007-1020; EMBO J, Aug; 35(16): 1766-1778). We performed PI staining in neutrophils from HV and SLE patients. Cells were isolated and seeded on confocal dishes (coated with Poly-D-lysine) for 30 mins for stable adhesion. Then cells were stained with PI.

The results showed significantly increased PI staining in neutrophils from SLE patients compared with neutrophils from HV (Fig R11).

We also isolated bone marrow neutrophils from WT and *Gsdmd*^{-/-} mice. Cells were then treated with IFN- γ plus LS treatment for 12 h. The results showed that IFN- γ plus LS treatment increased PI staining in neutrophils, which was significantly suppressed in *Gsdmd*^{-/-}-deficient cells (Fig R11). The results indicated deficiency of GSDMD significantly suppressed pore formation and PI staining.

Fig R11. (A) Confocal analysis of PI staining. Peripheral blood neutrophils were isolated from HV and SLE patients. Cells were seeded on confocal dishes and stabilized for 30 mins. (B) Quantitative analysis of the percentage of PI-stained cells. Each field of view represents one HV or SLE patient. n = 6 HV or SLE patients per group from 3 independent experiments. (C) Confocal analysis of PI staining. WT and *Gsdmd*^{-/-} murine bone marrow neutrophils were isolated. Cells were treated with IFN- γ plus LS. (D) Quantitative analysis of the percentage of PI-stained cells. Each field of view represents one mouse. n = 6 biological replicates and 3 independent experiments. (Added in Fig S2, A-D).

5. Fig 3. This referee may have missed this, but were the reverse controls for the “we isolated neutrophils from HVs and cultured these cells with the serum of SLE patients”

done so neutrophils from SLE cultured with serum from HV? If the pathology relies upon upregulation of GSDMD, caspase 1 and caspase 4 its not completely clear why the HV neutrophil plus SLE serum experiment works?

Response: Thank you for the remarkable suggestion. As shown in Fig 3A, peripheral blood from HVs was also primed with IFN- γ , we have corrected the obscure description in this sentence. Before, we performed experiments with neutrophils from SLE patients and cultured them with serum from HV. Neutrophils from SLE patients did not undergo NETs after treatment of serum from HVs. However, neutrophils from SLE patients were sufficient to undergo NETs after treatment of serum from SLE patients and without IFN- γ priming (Fig R12, A and B). Furthermore, we also isolated low density granulocytes (LDGs) from SLE patients. The results indicated spontaneous release of NETs in LDGs, which was also suppressed by DSF, but not GSK484 (Fig R12, C and D). Inhibition of GSDMD by DSF significantly suppressed NETs in LDGs. These results fully confirmed that GSDMD is required for NETs in neutrophils from SLE patients after LS treatment or spontaneous NETs of LDGs. Our data also showed the level of GSDMD, caspase-1, and caspase-4 in human neutrophils from HV and SLE patients (Fig 1O and Fig 5K), which showed an increased activation of GSDMD, caspase-1, and caspase-4 in neutrophils from SLE patients. From this perspective, we concluded that IFN- γ and LS promotes the activation of caspase-1/4/GSDMD pathway.

Fig R12. (A) Confocal analysis of SG, TOMM20, and Mitotracker staining in peripheral blood neutrophils isolated from SLE patients. Cells were pretreated with DSF, followed by treatment with LS. (B) Quantitative analysis of the percentage of released DNA. Each field of view represents one SLE patient. n = 6 SLE patients from 3 independent experiments. (C) Confocal analysis of SG, TOMM20, and Mitotracker staining in LDGs isolated from SLE patients. (D) Quantitative analysis of the percentage of released DNA. n = 6 SLE patients from 3 independent experiments. **(Only shown in the rebuttal).**

6. The statement “Gasdermin pores have lytic and non-lytic functions that control the release of intracellular contents” is not really accurate as the cell lysis is driven by NINJ1 so this phrase needs to be rephrased

Response: Thanks for your good suggestion, we have rephrased this statement as “Gasdermin pores have NINJ1-dependent lytic cell death and a non-lytic function that controls the release of pro-inflammatory cytokines.”

7. Fig 5 why has IFN now been introduced into the experimental protocol along with serum from lupus to cleave GSDMD? What is the IFN doing? Do the authors think IFN is altering GSDMD expression? The KO mice data in this experiment are not very convincing

Response: Thank you for your suggestions. In both Fig. 3 and 5, neutrophils from mice or humans were treated with IFN- γ . Since we identified both the activation of caspase-1 and caspase-11, treatment with IFN- γ may promote both canonical and noncanonical inflammasome pathways.

Caspase-1 and caspase-11 were upregulated by the stimulation of IFN- γ (Science, Sep; 341(6151): 1250-1253; Nature, Oct; 526(7575): 666-671), in which the precise mechanism was the upregulation of IFN-stimulated response elements (ISRE) and IFN-stimulated gene (ISG). Therefore, in our *in vitro* system, we used IFN- γ to prime neutrophils to increase the precursors of GSDMD, caspase-1, and caspase-11. A previous study also demonstrated that natural killer and T cells produce IFN- γ to prime caspase-11, which cleaves GSDMD to facilitate pyroptosis during *Burkholderia*

thailandensis infection (Cell Rep, Jul; 32(4): 107967), indicating the role of IFN- γ in driving caspase-11-dependent GSDMD cleavage and cell pyroptosis.

Finally, in lupus disease, the levels of IFN- γ in SLE patients and murine are significantly increased. IFN- γ plays an essential role in SLE (e.g., IL-18 production, innate and adaptive immune activation). Collectively, our data reveal an important role of IFN- γ in driving neutrophil pyroptosis in autoimmune diseases.

In response to your comments about KO mice data in Fig 5, we performed an immunoblot-based assay to examine the level of serpinb1, caspase-1, caspase-4, caspase-11, and GSDMD, which were included in Fig. 5E, 5G and 5I as representative data from three independent experiments. We have quantified these data in Fig. 5F, 5H and 5J, each symbol represents data from one mouse, data are pooled from 3 independent experiments.

8. Whats cleaving GSDMD upstream of caspase 1 or 11? What is the mechanism here by which Serpinb1 stops caspase 1 or 4 activation and GSDMD cleavage? Can the SLE pathology be ameliorated by an NLRP3 inhibitor for example if this is the protein responsible? The authors have quoted that ox-mitochondrial DNA has been suggested to be an activator of NLRP3 so it is unclear why this has not been investigated?

Response: Thank you for the important questions. In neutrophils of lupus disease, the upstream of caspase-1 and caspase-11 is serpinb1. Serpins are a superfamily of proteins that share a conserved tertiary structure, which was originally identified as an inhibitor of a serine protease (Biochemistry, Dec; 40(51): 15762-15770). In 2021, Jung et al. discovered that serpinb1 limited the activity of caspase-1, 4, 5, and 11 by suppressing their caspase-recruitment domain oligomerization and enzymatic activation. They also proved that *Serpinb1*^{-/-} neutrophils had consistently higher caspase-1 (FLICA-positive staining) than that of WT neutrophils (Fig R13). Besides, the expression of serpinb1 is the highest in neutrophils among other immune cells (www.immgen.org/Databrowser19). Our results fully confirmed the significant downregulation of serpinb1 in neutrophils from SLE patients and lupus mice. We also found that the expression of serpinb1 was significantly reduced after IFN- γ plus LS

treatment in bone marrow neutrophils.

Fig R13. Caspase-1 activation in *Serpinb1a*^{-/-} bone marrow neutrophils. FLICA-positive staining was analyzed by flow cytometry. (Adopted from *Nature immunology*, Mar; 20(3):276-287, Fig 4d).

Regarding your concern about whether SLE pathology is ameliorated by an NLRP3 inhibitor, we treated MRL/lpr mice with NLRP3 inhibitor MCC950 at 10 weeks. The results showed that MCC950 had a good protective effect in renal injury, but serum levels of autoantibodies were not suppressed by MCC950 (Fig.R6, in response to reviewer 1).

We also investigated the role of NLRP3 in GSDMD oligomerization. Bone marrow neutrophils were isolated from WT mice and cells were then pre-treated with MCC950. The results showed that MCC950 did not affect IFN- γ plus LS-induced GSDMD oligomerization (Fig R9, in response to reviewer 1), indicating that NLRP3 was not required for GSDMD oligomerization in neutrophils.

9. Is the Ox MtDNA controlled for LPS contamination? How did the authors ensure there was no LPS contamination from E coli?

Response: Human mtDNA was purified from 293T cells, and further amplified using PCR with an mtDNA-specific primer from the D-loop region. mtDNA was oxidized by UV-irradiation (250 mJ/cm²) according to previously published methods (*Immunity*, Sep; 39(3): 482-495). The concentration was relatively high, and about 2 μ L of mtDNA was added, which had little effect on the concentration of LPS. LPS, caspase-11, and GSDMD incubation systems were fully performed according to Shao Feng et al.

(Nature, Oct; 526(7575): 660-665). Purified LPS was purchased from Sigma-Aldrich (L4391), without *E. coli* contamination.

- Are there any flaws in the data analysis, interpretation and conclusions? - Do these prohibit publication or require revision?

See comments above

- Is the methodology sound? Does the work meet the expected standards in your field?

The methodology seems solid.

- Is there enough detail provided in the methods for the work to be reproduced?

The methodology description is reasonable.

Reviewer #3 (expertise in type-I interferons in autoimmunity, inflammation):

In experiments utilizing MRL.lpr mice, pristane-induced lupus (PIL), and peripheral blood cells obtained from human SLE patients, the authors report that gasdermin D (GSDMD) becomes oligomerized and activated resulting in inflammation and cell death. They report the effects of Gsdmd deficiency as well as after treatment of lupus mice with a non-selective small molecule inhibitor of GSDMD, disulfiram (DSF), in vitro and in vivo to show relevance. Observing beneficial effects, they propose that these inhibitors be considered for lupus therapy.

Inflammasome activation has been previously reported in murine models of SLE (reviewed in Kahlenberg & Kaplan, Curr Opin Rheumatol, 2014), and links between inflammasome activation, mitochondria, and cGAS-STING activation have been previously published (e.g. Aarenberg et al, Mol Cell, 2019). The most original aspects of the study are i) the experiments showing GSDMD oligomerization as a consequence of released oxidized mitochondrial DNA (mt DNA) which the authors show directly binds to GSDMD and ii) the beneficial effects of Gsdmd deficiency on murine lupus as well as the therapeutic effect of a non-selective inhibitor, disulfiram (DSF) in the two mouse models of lupus.

1. This is a very ambitious study resulting in much breadth but insufficient depth. Another general problem is that in many experiments only 3 samples are tested. The numbers should be increased to 5-6 in each group for most experiments.

Response: Thank you for your great comments and suggestions. We tried our best to broaden the depth of our manuscript in the revised version according to the reviewers and editors. In response to your concern that many experiments are only tested using three samples, as you suggested, all the experiments have been performed with at least five samples per group.

2. Much of the work focuses on the effects of lupus serum on mouse or human neutrophils in the generation of GSDMD-N. The manuscript could be strengthened by a more intense and thorough study of the lupus mice and spontaneous abnormalities in

human SLE cells.

Response: Thank you for your great suggestions. We have added more data on lupus mice and spontaneous abnormalities in human SLE neutrophils. We measured serum levels of mtDNA from PIL and *Gsdmd*^{-/-} mice (Fig R7, in response to reviewer 1). We also explored the role of DNAase, Ox-mtDNA, and NLRP3 inhibitor MCC950 in MRL/lpr mice and assessed these effects in the pathogenesis of lupus mice disease (Fig R6, in response to reviewer 1).

In neutrophils from SLE patients, we measured the level of GSDMD oligomerization using western blot analysis under non-reducing conditions. The results showed that the GSDMD oligomer was significantly increased in neutrophils from SLE patients, while no GSDMD oligomer was detected in neutrophils from HV (Fig R14).

Fig R14. (A) Western blot analysis of GSDMD oligomer under non-reducing conditions. Peripheral blood neutrophils were isolated from HV and patients with SLE. (B) Quantitative analysis of GSDMD oligomer/GAPDH. n = 6 HV or SLE patients per group pooled from 3 independent experiments. (Added in Fig 6, C and D).

We also performed PI staining in neutrophils from HV and SLE patients to further prove neutrophil cell death. Confocal analysis revealed increased PI staining in neutrophils from SLE patients compared with HV (Fig R11 A-D, in response to reviewer 2).

Moreover, we performed immunoprecipitation of GSDMD with TOMM20 or 8-OHdG in neutrophils from HV and SLE patients. The expression of TOMM20 and 8-OHdG was only detected in neutrophils from SLE patients (Fig R5), which indicated a

direct interaction of GSDMD and mtDNA in neutrophils from SLE patients.

3. Significant limitations of this study include a lack of rigor considering that there are too many experiments shown with single fluorescence or western blot without summary data to convince reviewers that ‘representative examples’ are reproducible findings. This is compounded by the fact that results in SLE patients are interwoven with the mouse data (often two different models) such that is difficult to ascertain the extent to which the paradigm applies to human SLE.

Response: Thank you for your suggestion, we have provided three independent experiments of each western blot and immunofluorescence analysis to strengthen the reproducibility of our data. All immunofluorescence and western blot data have been quantified and detailed information of quantification method has been provided in figure legend.

To the best of our knowledge, there is no mouse data that fully coincide with clinical human SLE disease. Two canonical and most widely used SLE mice models were used in this study. Pristane, a mineral oil (2,6,10,14-tetramethylpentadecane), can induce lupus-like disease in humans and mice characterized by immune complex nephritis with autoantibodies to single-stranded DNA, ribonucleoproteins, and overproduction of type I IFNs, similar to over half of patients with lupus (J Exp Med, Dec; 205(13): 2995-3006). However, the pristane-induced lupus model was an environmentally-induced lupus model that was not consistent with genetic factors-induced clinical SLE patients (Orthop Traumatol Surg Res, May; 96(3): 325). Meanwhile, the MRL/lpr strain is one of the best-established spontaneous models of SLE, in which lupus progression includes double-stranded DNA (dsDNA)-antibodies, proteinuria, vasculitis, crescent glomerulonephritis, and skin lesions (Autoimmun Rev, Sep;13(9): 963-73). The MRL/lpr strain has more anti-dsDNA antibodies and renal damage (crescent glomerulonephritis) than a pristane-induced mouse model. From the study of these two lupus models, we have provided a consummate and sufficient investigation of GSDMD in lupus disease. We think it is important to cover both human and mouse aspects in our study and we have made clearer statement when using mouse or human samples in the revised manuscript.

Fig. 1. The rationale for investigating the inflammasome pathway is not so clear since Fig. 1A shows very high upregulation of IRF3, IRF7 and Rel A with little or modest upregulation of Gsdmd and Casp 4. Please explain the relevance of IRF3, IRF7 and Rel A in these experiments.

Response: Thanks for your valuable suggestion. IRF3, IRF7, and Rel A are transcription factors implicated in the pathogenesis of SLE. IRF3 and IRF7-induced type I interferon production are induced by pathogen recognition receptors (PRRs) that identify pathogenic nucleic acids and regulate both antiviral and autoimmune responses. Previous studies have shown that serine/threonine kinase AKT2 interacts with IRF3 that attenuates IRF3-dependent type I interferon production in monocytes from SLE patients (EMBO J, Mar; 41(6): e108016). Through methylation quantitative trait loci (meQTL) analysis, lupus patients displayed an overlap with genetic risk loci for lupus including IRF7 (Ann Rheum Dis, Oct; 81(10): 1428-1437). Therefore, IRF3 and IRF7 are critical factors in SLE. The sample shown in Fig. 1A is a kidney from WT and pristane-induced lupus mice, which are specific models for type I interferon production manifesting in clinical SLE patients with higher type I interferon response. Therefore, IRF3 and IRF7 upregulation was expected, and our findings further indicate that IRF3 and IRF7 may be critical in the pathogenesis of lupus nephritis. The most abundant form of NF- κ B is NFKB1 complexed with the product of Rel A. Since NF- κ B regulates immune response, Rel A upregulation was expected.

Since IRF3, IRF7, and Rel A are transcription factors, the abundance of their mRNA level is relatively higher than a few proteins and enzymes. Although IRF3, IRF7, and Rel A were highly expressed in the kidney of lupus mice, their abundance was also high in the kidneys of control mice. Thereafter, we compared the fold change of these genes. Consequently, the fold change expression of gsdmd, caspase-1, and caspase-11 was higher than that of IRF3, IRF7, and Rel A (Fig. R10, in response to reviewer 2).

GSDMD is a downstream effector of both canonical inflammasome and non-canonical inflammasome pathways. However, the role of GSDMD downstream inflammasome in lupus disease is largely unknown. Of note, GSDMD promotes cell

pyroptosis, and the release of inflammatory cytokines (IL-1 β , IL-18) and DAMPs (IL-1 α , HMGB1, tissue factor, mtDNA). In this regard, understanding the role of cell pyroptosis and DAMP signaling is fundamental in shaping autoimmune diseases including SLE.

The low expression of a few proteins or enzymes does not mean they are unimportant. Besides, the activation of GSDMD, caspase-1, and caspase-11 depends on its cleavage, but not transcription level.

Fig. 1B. How do the authors exclude that GSDMD is more activated in macrophages than neutrophils? Figs. 1B-D and S1 B-D need rigorous quantification – interpretation of "representative figs" is not adequate.

Response: Thanks for your valuable suggestion. We performed flow cytometry to analysis the expression of GSDMD in peripheral blood neutrophils and macrophage precursor monocytes in SLE mice model. The results showed an increased expression of GSDMD in neutrophils from pristane-induced lupus mice compared with control. While the same level of GSDMD was observed in monocytes from pristane-induced lupus mice and control. Similar results were also observed in MRL/lpr mice. These results suggested that neutrophils are undergoing pyroptosis in lupus mice, but not monocytes (Fig. R15).

Fig R15. (A) Flow cytometry analysis of GSDMD in CD11b⁺Ly6G⁺ neutrophils and CD11b⁺Ly6G⁻ Ly6C⁺ monocytes. (B) Quantitative analysis of geometric mean of GSDMD in neutrophils in PIL mice. (C) Quantitative analysis of geometric mean of GSDMD in monocytes in PIL mice. (D)

Quantitative analysis of geometric mean of GSDMD in neutrophils in MRL/lpr mice. (E)
Quantitative analysis of geometric mean of GSDMD in monocytes in MRL/lpr mice. n = 6 mice per group pooled two independent experiments. (**Only shown in the rebuttal**).

We speculate that different cells play different roles in the pathogenesis of SLE. Although GSDMD was primarily expressed and focused on macrophages, the role of GSDMD in neutrophils in SLE patients remains unclear. We provide strong evidence on GSDMD activation in neutrophils from the peripheral blood of SLE patients and renal biopsy from lupus nephritis patients. Furthermore, we formed neutrophils-specific knockout of GSDMD in the lupus mice model, which displayed a significant protective effect on the pathogenesis of lupus disease. On the other hand, appropriate macrophage-specific knockout mice are unavailable, thus, it is difficult to confirm the role of GSDMD in macrophage of lupus disease. As a consequence, GSDMD activation in neutrophils is critical in SLE. Therefore, additional investigation is necessary to analyze the role of GSDMD on macrophage in SLE.

For your concern about Figs. 1B-D, we have provided rigorous quantification and data from three independent experiments.

What are the “Controls” in Fig. 1B-D?

Response: Thanks for your comments. All the data in Figure 1B, 1C and 1D have controls. In Figure 1B, the control group is normal kidney tissues from renal carcinoma patients. In Figure 1C, the control group is kidneys from WT mice after saline injection for 7 months. In figure 1D, the control group is MRL/mpj mice, which is a widely used control of MRL/lpr mice (J Exp Med, May; 189(10):1639-1648).

Fig. 1J. Why are two bands labelled GSDMD-N? Also seen in Fig. 5C & E.

Response: Thanks for your great suggestion. The two bands are actually the same one. The cleaved GSDMD-N is hard to expose when together exposed with pro-GSDMD. Here, the pro-GSDMD was covered with silver paper and further exposed the cleaved GSDMD-N, making it clear to visualize and compare the cleaved GSDMD-N from the indicated groups. We provide all the resource data of western blot in our revised paper.

To avoid misunderstandings, we have deleted the label of GSDMD-N in the blots with pro-GSDMD.

key Figure 2 G missing

Response: Thanks for your kind reminder, key Figure 2G was not missing, it is on the top right of the whole Figure 2.

Fig 3 & 4. These experiments essentially confirm other studies (see Chen, Demarco, Broz EMBOJ, 2019) indicating GSDMD can be responsible for NET formation, although in a lupus context.

Response: These experiments (Chen, Demarco, Broz EMBOJ, 2019) demonstrated that GSDMD regulates NETs. Nonetheless, whether GSDMD is a therapeutic target in SLE remains unreported. The most important role of neutrophils in SLE is NETs production, which significantly promotes autoantibody production, pDC activation, and renal damage. Further, we performed the release and oxidation of mtDNA by staining TOMM20 and 8-OHdG to further verify the role of GSDMD in mtDNA release. As a damage-associated molecular pattern, the release of mtDNA activates pDC and adaptive immune response in SLE. Thus, we believe that the results of Fig. 3 and 4 are necessary. These results are important in GSDMD-dependent NETs formation, and unravel a new role of GSDMD in mtDNA release in SLE.

In Fig. 3B, D, & L show differences in NET formation when GSDMD is lacking or inhibited. However, in most of these studies, there is not complete inhibition. The authors need to address this point - does this mean GSDMD contributes but does not fully explain NET formation? Fig. 3O, why was IFN-g added to the experiment? The red and purple are near impossible to see the overlap – which needs proper quantitation as mentioned for Fig. 1. The bottom panel needs to be labeled.

Response: We are grateful for your suggestion. Neutrophil death can transpire via diverse pathways. Whereas non-lytic forms of neutrophil death elicit an anti-inflammatory response, pathways ending with cell membrane rupture potentially

induce deleterious proinflammatory response and precipitate autoimmunity. Proinflammatory cell lysis death of neutrophils includes necroptosis, pyroptosis, and NETosis. Recent reports have identified that neutrophils mediate neutropenia in SLE (Nat Immunol, Sep;22(9):1107-1117).

According to our findings, inhibition or deletion of *GSDMD* does not completely block the formation of NETs, indicating other forms of neutrophil death including necroptosis or ferroptosis could also promote the formation of NETs.

As shown in Figures 5A and 5G, the effect of IFN- γ is promoting the expression of pro-GSDMD, pro-Caspase-1, and pro-Caspase-11 precursor, which is crucial for GSDMD-induced cell death.

We have changed the color from red and purple to green and red and counted the co-staining rate of 8-OHdG and TOMM20. We have also added bars in the enlarged pictures.

Fig. 4 E-G microscopy pictures don't always correspond to quantitation - DSF looks much less efficient at blocking 8OHdG in Fig 4E whereas DSF and MT look the same in Fig. 4G.

Response: Thanks for your comments. We believe these different quantifications were attributed to the different methods used for 8-OHdG detection. In Figure 4E, we performed immunofluorescence to detect 8-OHdG expression in cells. Consequently, no effect of 8-OHdG staining was observed in neutrophils, indicating that DSF does not affect mROS production. In Figure 4G, we used an ELISA method to detect 8-OHdG in the cultured medium, which indicates a release of damaged DNA. The released 8-OHdG was downregulated because DSF significantly inhibited GSDMD-dependent pore formation. Furthermore, n = 5 samples have been added in each experiment.

Fig. 5G, 4th lane looks more loaded than lane 5 (IgG depleted). This experiment needs proper quantitation (protein of interest / GAPDH ratio) over multiple experiments. Same with Fig. 4I – whereas in Fig. 5I the differences in serpin is clear (at least in this

one experiment).

Response: We have provided proper quantitation (relative protein expression/GAPDH). We also performed Figure 5G over multiple experiments. Proper quantitation (relative protein expression/GAPDH) was performed in Fig. 5I. We also performed Figure 5I over multiple experiments.

Fig. 6 contains necessary results for the conclusions yet only 3 data points are shown for key experiments – especially Fig. 6 G-L. The conclusions should be based on at least 5-6 mice per group. Fig. 6A, why is there no background in the *Gsdmd* KO lanes? What is the explanation for the discrepancy in the short term vs long term treatment with mitotracker? Fig. 6K is not adequately controlled. What about the addition of non-oxidized DNA. A dose titration effect would be more convincing.

Response: Thanks for your great suggestion. A total of 6 data points pooled from three independent experiments have been added in these key experiments (Fig. 6). We have shown another data in Fig. 6A, showing the background.

Fig R16 (A) Non-reducing western blot of GSDMD in bone marrow neutrophils from WT, Pristane and Pristane-*Gsdmd*^{-/-} mice. (B) Quantitative analysis of GSDMD oligomer/GAPDH. n = 5 mice per group and two independent experiments. **Added in Fig 6, A and B.**

The short-term treatment of mitoTEMPO was *in vitro* study in bone marrow neutrophils, which were pretreated before 2 h in IFN- γ plus LS stimulation. On the other hand, the long-term treatment was *in vivo* study in MRL/lpr mice. MitoTEMPO was continuously administered prophylactically to MRL/lpr mice for 7 weeks via the

subcutaneous pump, starting at 10 weeks of age (Nat Med, Feb; 22(2): 146-53).

For your concern on inadequate control of Fig. 6K. We added non-oxidized mtDNA as the control. Human mtDNA and Ox-mtDNA experiments were re-performed in LPS/Caspase-4/GSDMD system (Fig. R17, A and B). The results indicated that both non-oxidized mtDNA and Ox-mtDNA promote GSDMD oligomerization, and Ox-mtDNA increased GSDMD oligomer compared with non-oxidized mtDNA. We also performed a dose titration effect of Ox-mtDNA in GSDMD oligomerization. The results showed the oligomerization of GSDMD was on a dose-dependent manner of Ox-mtDNA (Fig. R17, C and D).

Fig. R17 (A) Purified GSDMD was incubated with LPS and caspase-4 *in vitro*. The system was then added with mtDNA (20 nM), and Ox-mtDNA (20 nM). Non-reducing Western blot of GSDMD oligomer. (B) Quantitative analysis of GSDMD oligomer/GSDMD monomer. n = 5 from 5 independent experiments. (C) Purified GSDMD was incubated with LPS and caspase-4 *in vitro*. The system was then added with different concentrations of Ox-mtDNA (0, 10, 20, 50 nM). (F) Quantitative analysis of GSDMD oligomer/GSDMD monomer. n = 5 from 5 independent experiments. **Added in Fig 6, M-P.**

Fig. 7C-D – transfection efficiencies of the constructs need to be shown by WB. Fig. 8 H-I need proper quantitation in at least 5 mice per group. Fig. 7, What does the Y axis “relative expression of IgG” mean? Fig. 7 - what was the result with H₂O₂ alone? Can statistical analysis be applied to a comparison between Figs E and F?

Response: Thanks for your suggestion. We have provided the transfection efficiencies of the constructs in Fig. s4 by Western blot (Fig R18).

Fig R18 (A-F) 293T cells were transfected with Flag-Full-GSDMD, Flag-GSDMD-N and Flag-GSDMD-C plasmid. Western blot of GSDMD (A), GSDMD-N (C) and GSDMD-C (E). Quantitative analysis of GSDMD (B), GSDMD-N (D) and GSDMD-C (F). (G-L) *Gsdmd*^{-/-} mouse

embryonic fibroblasts (MEFs) were transfected with Flag-GSDMD, Flag-GSDMD-N and Flag-GSDMD-C plasmid. Western blot of GSDMD (G), GSDMD-N (I) and GSDMD-C (K). Quantitative analysis of GSDMD (H), GSDMD-N (J) and GSDMD-C (L). n = 6 biological replicates pooled from 3 independent experiments (B, D, F, H, J and L). **Added in Fig S4, A-L.**

We have added the quantitative analysis of fig 8H and 8I, and the n was added to 5 mice per group.

In Figures 7A, 7C and 7D, GSDMD and IgG antibodies were used to pull down DNA. The D-loop of mtDNA was used for amplification. GSDMD pulldown primer cycle was relative to the IgG pulldown d-loop primer cycle. The protein-mtDNA immunoprecipitation was performed as previously described (Nat Microbiol, Mar; 2:17037).

We apologize for mislabeling of H₂O₂; the first group is no H₂O₂ stimulation. Once the H₂O₂ was added along, no cycle data was detected. Because our IP system was pulled down by the Flag antibody, which should be added at least one plasmid.

Figures 7E and 7F are the MST experiment, which cannot be quantified because of the fitting error; however, this experiment has been done independently for at least three times.

Fig. 8 The results from floxed GSDMD in neutrophils are an important addition to the experiments and help support the conclusions. The red and blue labeling in Figs 8C D-G seem opposite. The results in Fig. 8 C need better explanation in the figure legend – the green line above Gsdmd is higher in the knockout suggesting higher expression?

Response: Thanks for your valuable suggestion. We have relabeled Fig. 8 D-G and provided a better explanation in the Figure legend of Fig. 8C. In GSEA analysis, genes related to Interferon response or Inflammatory response are more enriched in WT mice, indicating downregulation of these pathways in the absence of GSDMD.

Fig. 9-Since MRL.lpr were treated at 6 weeks, this is more of a prevention than a treatment study. The results indicate partial improvement. IL-18 has been implicated in lupus pathogenesis in this strain and is also regulated by the inflammasome. What was

the effect of DSF on IL-18? Fig. 9K needs proper quantitation in at least 5 mice per group.

Response: We have performed the experiment of *in vivo* study of MRL/lpr mice, where the DSF was treated at 10 weeks as per the protocols in other studies (Nat Med, Feb;22(2):146-53). Please see the results in Fig R6 in response to reviewer 1. IL-18 levels have also been detected in serum from the indicated groups (Fig. R19). We also quantified Fig. 9K.

Fig R19 (A) ELISA analysis of IL-18 in serum from WT and *Gsdmd*^{-/-} mice. (B) ELISA analysis of IL-18 in serum from *Gsdmd*^{fl/fl} and *Gsdmd*^{fl/fl}*S100A8-Cre* mice. (C) ELISA analysis of IL-18 in serum from MRL/lpr and MRL/lpr with DSF-treated mice. n = 6 biological replicates and 2 independent experiments (Added in Figs S5H, S7D, and 9I).

Other points

Confusing nomenclature p.3 e.g. (Fig. S1, E and F), should read Fig. 1 E, F and Fig. S1)

Response: Thanks for your comments, we have corrected them.

l. 119 pyroptosis is not "new form of cell death" having been described more than 20 years ago

Response: Thanks for your comments, we have corrected them.

l. 151 the authors need to state when using mouse vs human

Response: Thanks for your comments, we have stated when using mouse vs human.

1. 177 DSF application in vitro needs viability controls

Response: Thanks for your suggestion. We have tested the role of DSF on the viability of bone marrow neutrophils from WT mice and peripheral blood neutrophils from HV. The results showed that DSF (5 μ M) does not affect the cell viability of bone marrow neutrophils (Fig R20A) and human neutrophils (Fig R20B).

R20. (A) CCK8 analysis of cell viability of bone marrow neutrophils treated with DSF at 5 μ M for different times. (B) CCK8 analysis of cell viability of peripheral blood neutrophils treated with DSF at 5 μ M for different times. n = 6 biological repeats and three independent experiments. (**Only shown in the rebuttal**).

1.233 too confusing to mix mouse and human especially when using human reagents on mice

Response: Thanks for your comments, we have done our best to clarify when using mouse or human samples in the revised manuscript.

1.284 - these are not typical ISGs

Response: Thanks for your comments, we have re-analysis the data of RNA-seq, and provided the data of canonical expression of ISGs.

1.301 looks like myeloid reduced

Response: This is because the pDC percentage increased from 5.43 to 11.6, hence the

previous group appeared to decrease, but our previous circle gate remained unchanged. Fig. S4 should have specific nephritis score also quantitation of IgG in the kidney. Sometimes the authors report proteinuria and sometimes UCAR. Kidney descriptions should be consistent

Response: Thanks for your suggestion. We have quantified Fig. S4 with a specific nephritis score and quantified IgG in the kidney (Fig S7, G and J). UCAR remained unchanged in pristane-induced mouse models because this model only had elevated proteinuria and normal renal function. The renal function index UCAR was significantly increased in the MRL/lpr model.

REVIEWERS' COMMENTS

Reviewer #1 (expert in SLE, renal disease, innate immunity in SLE):

The authors replied to all comments adequately and performed the necessary experiments - no more changes are needed.

Reviewer #2 (expert in gasdermins, innate immunity, inflammation):

The authors have addressed my concerns and provided new data to support their MS

Reviewer #3 (expert in type-I interferons in autoimmunity, inflammation):

The authors have improved their manuscript by attention to details of the critique. Overall, the results of experiments support their conclusions. There remain a few points that require attention.

Fig. R15 – the lack of increase in GSMD in monocytes (as compared to neutrophils) should be mentioned in the text (as not shown)

Fig, R20 – the result indicating that “DSF (5 μ M) does not affect the cell viability of bone marrow neutrophils (Fig R20A) and human neutrophils (Fig R20B)” should be mentioned in the main text (as not shown).

p.3 the authors should at least mention the increase in irf 3 & 7 expression in Fig 1A in the text as they are far more prominent than increased expression of Gsdm.

p.5 please explain why IFN-g was always added to experiments in the main text.

p.5 is the merge in Fig 3 supposed to show citH3 overlap with elastase? Wouldn't it be expected to be yellow?

Why is elastase needed - could you not just use stains for DNA & citH3 since this is an in vitro study with isolated neutrophils so elastase staining is unnecessary?

p.8. l.230 “Mito-TEMPO, following stimulation with LS plus IFN- γ resulted in decreased levels of oligomeric GSDMD (Fig.6, E and F), however, the levels of GSDMD-N remained unaffected (Fig. 6, G and H).” Why is oligomerization reduced but GSDM-N – the active form of GSDM unaffected?

p.8 l.248 Please explain how nuclear DNA and mtDNA in lupus serum may also promote GSDMD oligomerization.

pp. 10-11 and p.13. The authors state that “pDCs are the primary source of type I IFN in SLE patients.” However, the source of IFN-a in SLE has been challenged (e.g. PMID: 33262343 and PMID: 35583812) and it has also previously been shown by Reeves that IFN-a in the pristane induced lupus model is produced by monocytes, not pDC (PMID: 19047436). The authors should modify their comments accordingly. In addition, the authors use CD109 for detection of pDC. However, this marker is not specific for pDC. A more reliable marker is SIglec H (CD169).

Throughout the manuscript the authors use the terms like “suppressed” or “rescued” indiscriminately. When an intervention leads to return to wild type or baseline levels, the term rescued is appropriate whereas when the intervention causes a reduction that is statistically significant, the authors should use the term (statistically) significant reduction.

Discussion: It is not clear how the authors relate the main findings in their manuscript (GASDM induced release of oxidized mitochondrial DNA (a potent cGAS activator) and NETosis to the major mechanisms of lupus disease that they quote in the manuscript – that the IFN is induced by pDC, presumably by TLR. Perhaps a graphic abstract in the Supplement would help.

Fig. 7D fails to mention that IP was (presumably) performed prior to western blot.

RESPONSE TO REVIEWERS' COMMENTS

Reviewer #3 (expert in type-I interferons in autoimmunity, inflammation):

The authors have improved their manuscript by attention to details of the critique. Overall, the results of experiments support their conclusions. There remain a few points that require attention.

Fig. R15 – the lack of increase in GSDMD in monocytes (as compared to neutrophils) should be mentioned in the text (as not shown)

Response: Thanks for the valuable suggestions, we have provided the data of the expression of GSDMD in monocytes in the supplement data. We also mentioned this result in the main text.

Fig, R20 – the result indicating that “DSF (5 μ M) does not affect the cell viability of bone marrow neutrophils (Fig R20A) and human neutrophils (Fig R20B)” should be mentioned in the main text (as not shown).

Response: We thank the reviewer for the valuable suggestions. We have added the data of DSF on the cell viability of bone marrow neutrophils and human neutrophils in the supplement data. In addition, we mentioned these contents in the main text.

p.3 the authors should at least mention the increase in irf 3 & 7 expression in Fig 1A in the text as they are far more prominent than increased expression of Gsdmd.

Response: We appreciate the great suggestions. Since the increase of IRF 3 and 7 are far more prominent than the increased expression of GSDMD. We should provide these important results in the main text. Accordingly, we have added the description of the RNA-seq data of IRF 3 and 7 in the manuscript.

p.5 please explain why IFN-g was always added to experiments in the main text.

Response: Thanks for the great suggestions. We have added the reason for using IFN- γ in *in vitro* system to the main text.

p.5 is the merge in Fig 3 supposed to show citH3 overlap with elastase? Wouldn't it be

expected to be yellow?

Why is elastase needed - could you not just use stains for DNA & citH3 since this is an in vitro study with isolated neutrophils so elastase staining is unnecessary?

Response: The released NETs are decorated with citH3 and elastase, they are not necessarily to be overlapped with each other.

In response to your question regarding whether elastase staining is unnecessary. SG works well for staining extracellular DNA because it does not enter living cell. But at some point during cell death, the cells can be sufficiently permeabilized to allow for penetrance of SG. Therefore, it is worthwhile to measure histones as a second marker, or elastase and other neutrophil proteases (Myeloperoxidase, Cathepsin G). An optimal solution might be antibodies that detect DNA-histone complexes. Additional staining for citrullinated histones would enhance the results, so we used cit-H3. A third and important marker to determine neutrophils as a source of the DNA-histone complexes is taking NE measurements in the complexes (Nat Med. Mar 7;23(3):279-287.). So we think the staining of elastase is necessary.

p.8. 1.230 “Mito-TEMPO, following stimulation with LS plus IFN- γ resulted in decreased levels of oligomeric GSDMD (Fig.6, E and F), however, the levels of GSDMD-N remained unaffected (Fig. 6, G and H).” Why is oligomerization reduced but GSDM-N – the active form of GSDM unaffected?

Response: GSDMD-N is the amino-terminal pore forming domain of gasdermin D, which interacts with the plasma membrane and ~16 monomers oligomerize to form a gasdermin pore (Trends Cell Biol, Sep;27(9):673-684). Our results showed an increased level of GSDMD oligomer, and the molecular weight of this oligomer is 250 kD, indicating an octamer of GSDMD-N (31 kD*8=248 kD). And this increased GSDMD oligomer was reduced by Mito-TEMPO, suggesting a role of mROS in GSDMD oligomerization. This result is in consistent with a previous study showing the positive role of mROS in GSDMD oligomerization in macrophage (Cell, Aug 19;184(17):4495-4511.e19.).

The cleaved active form of GSDMD-N terminal was dependent on the enzyme that

cleave GSDMD, such as caspase-1 and caspase-11. Increased or reduced GSDMD-N does not mean they have a good state to form oligomer. And the oligomerization of GSDMD-N is dependent on mROS, but not the basic level of GSDMD. So the unaffected GSDMD-N means mROS does not influence the cleavage of GSDMD, but did affect its pore forming activity.

p.8 1.248 Please explain how nuclear DNA and mtDNA in lupus serum may also promote GSDMD oligomerization.

Response: Neutrophil phagocytosis can be activated by FcR (mBio, Oct 4;7(5):e01624-16) and Complement C3 (Adv Mater, Aug;34(34):e2203477). Since the FcR and Complement pathways are significantly activated in neutrophils in lupus, we speculate that nuclear DNA or mtDNA in serum may be phagocytosed by neutrophils. These nuclear DNA or mtDNA may directly interact with GSDMD to further promote GSDMD oligomerization. So when the lupus serum was treated with DNase I, there were less GSDMD oligomer.

pp. 10-11 and p.13. The authors state that “pDCs are the primary source of type I IFN in SLE patients.” However, the source of IFN- α in SLE has been challenged (e.g. PMID: 33262343 and PMID: 35583812) and it has also previously been shown by Reeves that IFN- α in the pristane induced lupus model is produced by monocytes, not pDC (PMID: 19047436). The authors should modify their comments accordingly. In addition, the authors use CD109 for detection of pDC. However, this marker is not specific for pDC. A more reliable marker is Siglec H (CD169).

Response: Thanks for the great comments. We have carefully read the papers as you suggested. Indeed, the exiting literatures are complex and sometimes contradictory. While many studies have suggested that endogenous nucleic acids forming immune complexes with autoantibodies as a stimulus for pDC activation (Nat Rev Immunol, Aug;8(8):594-606; J Immunol, Sep 15;171(6):3296-302; Arthritis Rheum, Aug;60(8):2418-27.), previous studies have reported both higher and lower number of pDCs in the blood (Science, Nov 16;294(5546):1540-3; Lupus, Jul;17(7):654-62.).

Unsorted PBMCs from SLE patients have been shown to produce lower levels of IFN- α in response to TLR9 stimulation, while other studies reported enhanced TLR7 mediated IFN- α production by pDCs of SLE patients (Arthritis Res Ther, 10(2):R29; Arthritis Res Ther, 19;19(1):234.). So it is not clear whether any alteration in the pDC phenotype is the result of chronic inflammation or therapy, nor what underlying mechanism determines their function in the early stages of human autoimmunity.

In the studies you have mentioned, the major producers of type I interferon were found to be non-immune cells in skin biopsy samples from patients with cutaneous lupus erythematosus (PMID: 33262343). In another study, pDC numbers are reduced in preclinical autoimmunity and SLE, which has impaired function of type I interferon production (PMID: 35583812). And in the pristine induced lupus model, the source of IFN- α is monocyte (PMID: 19047436).

To be accurate, we delete the sentence “pDCs are the primary source of type I IFN in SLE patients.”

Regarding your second good suggestion about the specific cell marker of pDC. We realized that the specific cell marker for pDC is B220⁺SiglecH⁺CCR9⁺. To be more accurate, we delete the results of flow cytometry data of pDC.

Throughout the manuscript the authors use the terms like “suppressed” or “rescued” indiscriminately. When an intervention leads to return to wild type or baseline levels, the term rescued is appropriate whereas when the intervention causes a reduction that is statistically significant, the authors should use the term (statistically) significant reduction.

Response: Thanks for your great suggestion, we have corrected the inappropriate descriptions in the result part.

Discussion: It is not clear how the authors relate the main findings in their manuscript (GSDM induced release of oxidized mitochondrial DNA (a potent cGAS activator) and NETOsis to the major mechanisms of lupus disease that they quote in the manuscript – that the IFN is induced by pDC, presumably by TLR. Perhaps a graphic abstract in the

Supplement would help.

Response: Thanks for the great suggestions. We have provided a graphic abstract in Figure 9q.

Fig. 7D fails to mention that IP was (presumably) performed prior to western blot.

Response: Thanks for the great suggestions. The IP was performed prior to Western blot. The cell lysis was firstly incubated with GSDMD, then the protein A/G beads were added to pull down GSDMD. Then the pulled protein was performed on Western blot. We have mention the IP in the figure legend.